



# Comparison of three DWM-based wake models at below-rated wind speeds

Øyvind Waage Hanssen-Bauer[1], Paula Doubrawa[2], Helge Aa. Madsen[3], Henrik Asmuth[4],
Jason Jonkman[2], Gunner C. Larsen[3], Stefan Ivanell[4], and Roy Stenbro[1]

[1]Institute for Energy Technology
[2]National Renewable Energy Laboratory
[3]Technical University of Denmark
[4]Wind Energy Division, Department of Earth Sciences, Uppsala University, Visby, Sweden

**Correspondence:** Øyvind Waage Hanssen-Bauer (oyvind.hanssen-bauer@ife.no)

**Abstract.** Wind turbine wake models are essential tools for predicting power losses and structural loads in wind farms. Among them, the dynamic wake meandering (DWM) model, included as a recommended approach in the International Electrotechnical Commission design standard, is a widely used engineering-fidelity method that balances accuracy and computational cost. This study compares the performance of three DWM-based wake model implementations (from the Technical University of Denmark, the National Renewable Energy Laboratory, and the Institute for Energy Technology) under below-rated wind speed conditions. Model predictions of wake flow, power output, and structural loads for a four-turbine row are evaluated across different ambient turbulence levels and wind-direction misalignments, and compared against high-fidelity large-eddy simulation results. All three models captured the overall wake evolution and mean turbine performance with reasonable accuracy; predicted time-averaged thrust and power were typically within 5—10 % of the large-eddy simulation benchmark. However, notable differences emerged in wake structure and unsteady load predictions, with discrepancies increasing for turbines further downstream. These differences highlight the importance of modelling choices such as wake summation and turbulence treatment, which strongly influence power-deficit and fatigue-load predictions. Comparison with large-eddy simulations reveals the strengths and weaknesses of each approach, indicating where improvements are needed. In general, the findings suggest directions for refining DWM models and improving their fidelity, ultimately enabling more robust wake predictions for wind farm design and operation.

## 1 Introduction

The wind energy industry has undergone significant development in recent decades, evolving from isolated, low-efficiency turbines to large-scale, modern wind farms. In these farms, spatial constraints and the need to minimize infrastructure and maintenance costs often lead to farm layouts with tightly spaced turbines. This evolution has increased the focus on turbine–turbine interactions, as wake effects have been identified as a major contributor to energy losses and elevated structural loads throughout the farm.





To maximise energy yield, the industry commonly employs simplified engineering models for steady-state wake prediction during design and operational planning. However, wakes from upstream turbines not only reduce wind speeds but also generate unsteady turbulence, which impacts the performance and fatigue-loading of downstream machines. Because steady-state models are inherently unable to capture these unsteady flow phenomena, they are not suitable for load assessments. Instead, the industry commonly relies on the effective turbulence model (International Electrotechnical Commision, 2019) for structural load calculations, which does not simulate individual wakes explicitly but approximates their impact by artificially increasing the ambient turbulence intensity. The alternative approach to consider wake effects on turbine loads according to international wind turbine design standards is the dynamic wake meandering (DWM) model (Larsen et al., 2008). This approach explicitly simulates individual wakes as convecting, meandering flow fields, where the velocity-deficit is advected downstream with stochastic lateral and vertical motion driven by ambient large-scale turbulence, superimposed on an ambient wind field. By capturing key unsteady wake dynamics such as meandering and advection, DWM-based models include physical phenomena that are absent from simpler steady-state models, yet remain orders of magnitude more computationally efficient than high-fidelity large-eddy simulations (LES). Recent work by Doubrawa et al. (2023) showed that even though on average the effective turbulence model and the DWM model predict similar intra-farm flow characteristics and, when coupled with aeroelastic solvers, turbine structural loads, much more insight and direction variability arises from the DWM model that the effective turbulence model cannot resolve. DWM models enable realistic load predictions under wake conditions — an essential capability for wind farm design and certification.

Since its introduction in the early 2000s, the DWM model has undergone continuous refinement. Several research groups have proposed enhancements or modifications to the original formulation, including alternative meandering algorithms, variations in wake-deficit shapes (Doubrawa et al., 2017; Branlard et al., 2023), improved wake superposition techniques (Machefaux et al., 2016; de Vaal and Muskulus, 2021), and more advanced treatments of wake-added turbulence (Madsen et al., 2005; Keck et al., 2015; Branlard et al., 2024). These efforts have led to a range of DWM-based implementations, such as the original model integrated with DTU's aeroelastic software HAWC2, NREL's FAST.Farm tool (Jonkman et al., 2017), and the more recent WIFET wake model (Hanssen-Bauer et al., 2020; de Vaal and Muskulus, 2021), each incorporating unique sub-models. While grounded in the same core physical principles, their predictions can differ substantially due to implementation choices.

DWM-based models have been calibrated and compared with high-fidelity large-eddy simulations coupled with LES actuator-line turbine models (LES-ALM) (Madsen et al., 2010; Jonkman et al., 2018; Doubrawa et al., 2018; Shaler and Jonkman, 2021; Hanssen-Bauer et al., 2020) and also validated with full-scale field measurements (Madsen et al., 2010; Larsen et al., 2013, 2015, 2017). Yet direct intercomparisons between different DWM implementations remain limited. A notable exception is the benchmarking study by Asmuth et al. (2022), which compared six numerical models—including DWM implementations from DTU and NREL, and the LES-ALM software Ellipsys3D—with full-scale measurements from the DanAero experiment. That study focused on a two-turbine setup under below-rated wind conditions, analysing one full-wake and one partial-wake case. However, the scope was limited to the response of two turbines and the wake flow behind only the upstream rotor, leaving the effects of multiple interacting wakes unexamined.



Our recent comparison of DWM-based models extended the benchmarking to an above-rated wind speed case, involving a four-turbine row aligned with the incoming wind and a single ambient turbulence condition (Hanssen-Bauer et al., 2023). That study revealed substantial discrepancies between the model implementations. While time-averaged wake-deficits and power outputs were generally consistent across models and in reasonable agreement with LES, fatigue-load predictions diverged significantly further downstream, with differences reaching up to 25 % of reference values. These results underscore how implementation details, such as wake-merging methods and turbulence modelling, can critically affect load predictions, even under otherwise comparable conditions. They also highlight the need for continued evaluation and improvement of engineering-fidelity wake models before they can be fully relied upon in design and certification workflows.

In the present study, we extend the earlier above-rated comparison to systematically evaluate three DWM-based wake models under below-rated wind speed conditions, while introducing two further variables: ambient turbulence intensity and wind-direction misalignment. Specifically, we analyse three inflow conditions representative of low to moderately high turbulence environments and two wind alignment scenarios — one with flow aligned with the turbine row, resulting in a full-wake configuration, and another with a small offset angle introducing a partial-wake condition. A high-fidelity LES-ALM is used as the reference benchmark, following the methodology of our previous study (Hanssen-Bauer et al., 2023). This setup enables an in-depth assessment of wake evolution, power production, and structural load indicators along a row of turbines for each DWM model, across all combinations of wind speed, turbulence, and alignment.

The primary objectives of this study are twofold: (1) to evaluate each DWM model's accuracy relative to LES predictions, identifying the deviations in wake behaviour and turbine fatigue response; and (2) to investigate how differences in sub-modelling strategies — such as wake meandering formulations, velocity-deficit profiles, multi-wake superposition methods, and wake-added turbulence treatments — affect model performance. By isolating and analysing these factors, we aim to explain the observed differences and identify the most influential modelling assumptions, thereby informing future development of accurate, robust engineering-fidelity wake models for wind farm applications.

## 2 Methodology

In this study, we compare three different DWM-based wake models with high-fidelity LES-ALM. The original DWM model developed at the Technical University of Denmark (DTU) is referred to as $DWM_{DTU}$. The second DWM model uses the National Renewable Energy Laboratory (NREL) DWM implementation in FAST.Farm, named $DWM_{NREL}$ in this study. The third model, named $DWM_{IFE}$, uses the DWM implementation WIFET Farm from the Institute for Energy Technology (IFE). This model is newly developed in the NEXTFARM project (RCN, 2025) and is an extension to the aeroelastic tool 3DFloat (Nygaard et al., 2016). The LES-ALM simulations were performed by Uppsala University and are hereafter called $LES_{UU}$.

### 2.1 Test cases

In this study, we consider the same simple farm layout as in Hanssen-Bauer et al. (2023), a row of four NREL 5-MW reference turbines (Jonkman et al., 2009) spaced 7.5 diameters (7.5D) apart. The NREL 5-MW turbine has a rotor diameter of D = 126



m, a hub height of 90 m, a rated speed of $11.4 \mathrm{~ms}^{-1}$, and a rated aerodynamic power of 5.3 MW. All numerical models, both DWM and LES, use the same incoming wind field, the LES-generated precursor described in Sect. 2.3. In that way we exclude

the effect of different inflow models, and enable to investigate the differences in the wake models and their isolated impact on power and fatigue-loads. However, an important exception is the computation of the meandering in the $\mathrm{DWM_{DTU}}$ model, which is derived from a Mann turbulence box with a grid size of one diameter (Madsen et al., 2008, 2010). As this approach is an integrated part of the model and its calibrated model parameters, it was found necessary not to deviate from this.

Three wind fields with varying ambient turbulence intensity ($\mathrm{TI_a}$) were generated, representing low, medium, and high

turbulence inflow conditions. Table 1 gives details about the flow at hub height for the different cases. While the aim was to have three wind boxes with identical below-rated mean wind speed at hub height, we see that in fact the mean wind speed varies. For the highest $\mathrm{TI_a}$, the mean wind speed is close to, but still below, the rated wind speed. The inflow data provided to the DWM models was sampled in a separate precursor run of the main LES without turbines in a plane 1D upstream of the position of the most upstream turbine, hereafter referred to as turbine 1. This way, it is ensured that the inflows seen by the

turbines are as similar as possible. For the DWM simulations, the LES-generated wind field was then imposed 1D upstream of turbine 1, and the simulations were run for 52.5 min. To exclude any transient effects at the beginning of the simulations, the first 7.5 min were excluded from the results, resulting in an effective simulation length of $t_{\mathrm{sim}} = 45$ min. This corresponds to $6.7 L_x/U_\infty \leq t_{\mathrm{sim}} \leq 8 L_x/U_\infty$ for the different cases, where $L_x$ is the longitudinal length of the flow regime, and $U_\infty$ is the mean undisturbed ambient wind speed.

In total, four simulation cases were run in this study. For three of the cases, the mean wind-direction was in line with the row of turbines but with varying inflow conditions. Here, the turbines downstream of turbine 1 were operating in fully waked conditions. The fourth case was run with medium ambient turbulence conditions, but with an offset angle of $5°$ between the mean wind-direction and the turbine row, resulting in a case where turbines 2–4 operated under partially waked conditions. However, as for all cases in this study the rotors were aligned with the mean wind-direction, i.e. no yaw misalignment relative to

the mean wind occurred. Due to an error in the setup of the LES-ALM simulation of the first case, the rotor was run with $0°$ tilt angle, and not the correct $5°$ tilt angle of the NREL 5-MW turbine. As the LES-ALM simulations are rather computationally expensive, it was decided to keep a $0°$ tilt angle for the first case and adjust the DWM simulations accordingly, while for the remaining simulations the tilt angle was adjusted to $5°$ (see table 1).

As in Hanssen-Bauer et al. (2023), the turbines were forced to operate at fixed rotor speeds and blade pitch angles in all

simulations. These predefined values were set by first running the $\mathrm{DWM_{IFE}}$ and $\mathrm{DWM_{NREL}}$ models with variable rotor speeds and blade pitch with the same inflow and subsequently using the mean of the time-averaged values from the two runs for the final simulations. The resulting rotor speeds are given in table 1, while the blade pitch angles were $0°$ for all turbines, as expected for below-rated conditions. The approach for $\mathrm{DWM_{DTU}}$ is to run all upstream wake-generating turbines at free inflow conditions, except the turbine where the loads are simulated. So when the loads of turbine 4 were simulated, turbines 1 to 3

were set to the rotor speeds given for turbine 1 in table 1, while turbine 4 was set to the RPM specified for that turbine.

To get results comparable to the ALM in $\mathrm{LES_{UU}}$, it was decided to run the aeroelastic solvers coupled to the DWM wake models with rigid rotors and exclude all the effects from the tower. Aerodynamic forces, including gravity forces, along the



**Table 1.** Inflow conditions at hub height, resulting pre-defined RPM values, and rotor tilt angle for the simulation cases

|  | $U_{\text{hub}}$ [ms$^{-1}$] | TI$_{\text{hub}}$ [%] | RPM turbine 1 [min$^{-1}$] | RPM turbine 2 [min$^{-1}$] | RPM turbine 3 [min$^{-1}$] | RPM turbine 4 [min$^{-1}$] | Rotor tilt [°] |
|---|---|---|---|---|---|---|---|
| Low TI$_{\text{a}}$ | 8.86 | 4.6 | 10.23 | 8.43 | 8.36 | 8.35 | 0 |
| Medium TI$_{\text{a}}$ | 8.98 | 8.8 | 10.51 | 8.76 | 8.57 | 8.54 | 5 |
| High TI$_{\text{a}}$ | 10.63 | 12.0 | 11.86 | 10.76 | 10.44 | 10.38 | 5 |
| Medium TI$_{\text{a}}$/ Skewed inflow | 8.98 | 8.8 | 10.36 | 9.40 | 9.32 | 9.27 | 5 |

radial span of the blade were reported from all simulations, and power and loads were calculated from these forces using the same algorithms. This is the same procedure used in Hanssen-Bauer et al. (2023).

To compare fatigue-damage calculations using the different wake models, 45-min damage equivalent loads (DEL) were calculated. Based on the Palmgren–Miner damage-summation rule with Goodman's correction, a DEL is a load that, at a chosen equivalent number of cycles—here $N_{eq} = 45 \cdot 60 = 2700$ (i.e. a load at 1 Hz for 45 min)—will give the same damage as the summation of the $k$ cycles $N_k$ of load ranges $S_k^m$ determined using rainflow counting (Rychlik, 1987):

$$\text{DEL} = \left( \frac{1}{N_{eq}} \left( \sum_{k=1}^{N_k} N_k S_k^m \right) \right)^{\frac{1}{m}} \tag{1}$$

**2.2 The DWM models**

The original DWM model is based on the assumption that the quasi-steady wake-deficit, obtained from a thin shear-layer approximation of the Navier-Stokes equations, meanders in a stochastic manner due to the large-scale turbulent structures in the wind, and that the self-generated turbulence field in the wake can be superimposed onto the wake-deficit and exposed to the same dynamics. In this study, we compare three DWM-based wake models from DTU, NREL, and IFE. An overview of

the differences between these three DWM model implementations are given in Hanssen-Bauer et al. (2023). What follows is a summary of the most important differences needed to understand the discrepancies in the results.

**2.2.1 Initial velocity profile**

DWM$_{\text{DTU}}$ and DWM$_{\text{NREL}}$ obtain the initial velocity profile behind the turbine from the blade element momentum (BEM) model (Madsen et al., 2008, 2010), but the wake profile is adjusted by including a simple closed-form modification taking care

of the pressure recovery in the wake near field. DWM$_{\text{IFE}}$ on the other hand, assumes a Gaussian wake-deficit profile for all positions downstream of the turbine, and the initial wake centre deficit is obtained from $C_T(U)$ tables with thrust coefficient as function of wind speed for the specific turbine.





### 2.2.2 Thin shear-layer approximation and eddy viscosity model

All the three DWM implementations build on the same assumption of an axisymmetric wake with a thin shear-layer approxi-
mation of the Navier-Stokes equations, where the pressure term is neglected. As a turbulence closure of the equations, an eddy
viscosity model consisting of two terms, is applied. The first term models the contribution related to the ambient wind shear and
scales with the turbulence intensity, while the second term is related to the wake shear. The model includes filter functions to
adjust the model in the near-wake region where the assumption of negligible pressure variations is not valid. The details of the
eddy viscosity model, with the associated filter functions and calibration constants, vary between the DWM implementations
(for details, see Madsen et al., 2010; Jonkman et al., 2017; de Vaal and Muskulus, 2021).

### 2.2.3 Wake transport velocity

The wake-deficit is transported downstream by the wind, but since the free stream velocity is disturbed by this deficit itself,
the choice of wake transport velocity is not trivial. While $\text{DWM}_{\text{DTU}}$ applies a transport velocity of $U_\infty$, $\text{DWM}_{\text{IFE}}$ uses the
approximation of $0.8U_\infty$, estimated by Keck et al. (2013). $\text{DWM}_{\text{NREL}}$, on the other hand, is calculating the local velocity at the
position of each wake slice, which varies in both time and space; therefore, the wake accelerates from near-wake to far-wake,
because the wake-deficits are stronger in the near-wake and weaken downwind.

### 2.2.4 Wake summation

For multiple wake situations, where a turbine's incoming flow field is affected by more than one wake of upstream turbines,
$\text{DWM}_{\text{DTU}}$ distinguish between below- and above-rated wind speed conditions (Larsen et al., 2015):

$$
\quad U_w(x,y,z) = U_\infty - \begin{cases} \max_i \left( U_\infty - u_w^i(x,y,z) \right), & U_\infty \leq U_{\text{r}} \\ \sum_i (U_\infty - u_w^i(x,y,z)), & U_\infty > U_{\text{r}} \end{cases} \tag{2}
$$

Here, $U_\infty$ is the undisturbed free-stream velocity, $u_w^i$ is the wake velocity induced by turbine $i$, and $U_{\text{r}}$ is the turbine's rated
wind speed. In this study the wind speed is always below rated, therefore the upper expression is used. The maximum deficit
operator looks at the meandered wake-deficit from each upstream turbine when operating in isolation (i.e., experiencing free-
stream velocity), and assumes that the total incoming wake-deficit can be approximated to be the maximum single wake-deficit,
evaluated at each radial position of the turbine of interest.

In $\text{DWM}_{\text{NREL}}$ the axial velocity-deficits are superimposed using a local root-sum-square method, where the wake of each
individual turbine is calculated using the local incoming wind velocity of that turbine, meaning that the wakes are calculated
in a sequential way from upstream to downstream (Jonkman et al., 2017):

$$
U_w(x,y,z) = U_\infty - \sqrt{\sum_i (u_0^i - u_w^i(x,y,z))^2} \tag{3}
$$





Here, $U_\infty$ is again the undisturbed free-stream velocity, $u_0^i$ is the local incoming wind velocity of turbine $i$, and $u_w^i$ is the wake velocity induced by turbine $i$.

Radial velocity-deficit fields are superimposed using a linear summation method, where the wake of each individual turbine is calculated in the same sequential way as for the axial component.

$\mathrm{DWM_{IFE}}$ uses the momentum conserving summation method derived by Zong and Porté-Agel (2020a) for wake superposition. This is an iterative method, where the velocity-deficits from the upstream turbines are summed weighted on the ratio of the mean convection velocity of the individual wake, $u_c^i(x)$, and the convection velocity of the combined wakes, $U_c(x)$:

$$U_w(x,y,z) = U_\infty - \sum_i \frac{u_c^i(x)}{U_c(x)}(U_\infty - u_w^i(x,y,z)) \tag{4}$$

$$u_c^i(x) = \frac{\iint u_w^i(x,y,z) \cdot (u_0^i - u_w^i(x,y,z))dydz}{\iint (u_0^i - u_w^i(x,y,z))dydz} \tag{5}$$

$$U_c(x) = \frac{\iint U_w(x,y,z) \cdot (U_\infty - U_w(x,y,z))dydz}{\iint (U_\infty - U_w(x,y,z))dydz} \tag{6}$$

### 2.2.5 Tilt and yaw misalignment

The implementations by $\mathrm{DWM_{DTU}}$ and $\mathrm{DWM_{IFE}}$ used in this study do not take into account any impact on the flow due to tilt and yaw misalignment between the rotor and the flow. However, in the latest version of $\mathrm{DWM_{DTU}}$ a model to account for flow effects due to yaw misalignment using a Hills vortex analogy (Larsen et al., 2020) the implementations by $\mathrm{DWM_{DTU}}$ and $\mathrm{DWM_{IFE}}$ used in this study do not take into account any impact on the flow due to tilt and yaw misalignment between the rotor and the flow. In $\mathrm{DWM_{NREL}}$, tilt and yaw misalignments are accounted for and affect wake deflection (Jonkman et al., 2017). The wake planes in the $\mathrm{DWM_{NREL}}$ model are oriented by the rotor centreline and not the wind-direction, causing the wake to deflect based on tilt and yaw misalignment because the wake-deficit normal to the rotor introduces a velocity component that is not parallel to the incoming flow. $\mathrm{DWM_{NREL}}$ has in addition a newly implemented curled-wake model with improved accuracy for large rotor misalignments (Branlard et al., 2023), but this model is not used in this study.

### 2.2.6 Ground effects

$\mathrm{DWM_{NREL}}$ does not yet have a model to account for ground effects on the flow field. For both $\mathrm{DWM_{DTU}}$ and $\mathrm{DWM_{IFE}}$ such a model is implemented, but not used for the simulations in this study. For $\mathrm{DWM_{IFE}}$'s part, the mirroring model to handle ground effects was used in the simulations in Hanssen-Bauer et al. (2023), but it was later seen that this gave unrealistically high deficits close to the ground, and $\mathrm{DWM_{IFE}}$ performed better when turning off this model.





### 2.2.7 Wake-added turbulence and turbulence build-up

Wake-added turbulence is the self-generated small-scale turbulence in the wake of the turbines due to wake shear and the breakdown of the wake tip vortices, and comes in addition to the conventional atmospheric boundary layer turbulence. $\text{DWM}_{\text{DTU}}$ is the only DWM implementation including a wake-added turbulence model in the simulations performed in this study. At an early stage in the development of the DWM model at DTU, it became clear from the comparison of model simulations with detailed inflow measurements on a full-scale turbine with angle of attack and relative velocity to a blade section, that additional turbulence to what is generated from wake meandering was necessary to model (Madsen et al., 2005). In practice the wake self-generated turbulence, which is of particular importance in case of stable stratification of the atmospheric boundary layer, is modelled based on an isotropic Mann box with smaller length scale [1] than the conventional inflow turbulence, and transformed into an inhomogeneous turbulence field by a scaling factor $k_{mt}$ varying radially based on the wake-deficit strength and the wake shear-layer velocity gradient:

$$k_{mt} = \left| \frac{u_w(r)}{U_\infty} \right| k_{m1} - \left| \frac{\partial (u_w(r)/U_\infty)}{\partial r} \right| k_{m2}. \tag{7}$$

Here $k_{m1} = 0.6$ and $k_{m2} = 0.35$ are empirical factors tuned by comparison with inflow and load measurements on a full scale turbine (Madsen et al., 2008) and with actuator-line simulations (Madsen et al., 2010). Later, an improvement to the original model to account for the build-up of turbulence inside a wind farm was suggested (Keck et al., 2015), but this is not included in the $\text{DWM}_{\text{DTU}}$ model.

The results by $\text{DWM}_{\text{NREL}}$ are not including any wake-added turbulence model in this study. However, an improved wake-added turbulence model has recently been implemented in FAST.Farm (Branlard et al., 2024).

$\text{DWM}_{\text{IFE}}$ does not include a wake-added turbulence model for load calculations analogous to the one formulated in the original DWM model. However, the increased TI in the wake due to the turbulence-generating wake-deficit shear is modelled based on the eddy viscosity formulation in the wake-deficit model, and the total contribution of increased TI from all upstream wakes is estimated by a root-sum-square summation (de Vaal and Muskulus, 2021). Thus, the increased effective TI felt by a turbine operating under waked conditions is taken into account and affects the development of its own wake downstream.

### 2.2.8 Aeroelastic solvers

All DWM models are coupled to an aeroelastic solver for calculating blade forces. $\text{DWM}_{\text{DTU}}$ is coupled to HAWC2 (Madsen et al., 2020), $\text{DWM}_{\text{NREL}}$ to OpenFAST (NREL, 2025), and $\text{DWM}_{\text{IFE}}$ to 3DFloat (Nygaard et al., 2016). In all these aeroelastic solvers the blade forces are obtained from BEM, although in different implementations, with Prandtl blade tip correction (Glauert, 1935). $\text{DWM}_{\text{NREL}}$'s OpenFAST has in addition a blade root correction.

---

[1] $L = D/8$, where $L$ is the length scale of the spectral velocity tensor and $D$ is the turbine diameter, opposed to $L = 33.6$ m which is recommended for atmospheric turbulence above 60 m (International Electrotechnical Commision, 2019).





## 2.3 Large-eddy simulations

The LES-ALM reference case, $LES_{UU}$, and the three inflow wind fields used by all the numerical models in this study, is
computed using the numerical framework EllipSys3D (Michelsen, 1994a, b; Sørensen, 1995), and is identical to the solver used in our recent comparison at above-rated wind conditions (Hanssen-Bauer et al., 2023). The solver was also participating in the mentioned benchmarking study against full-scale measurements (Asmuth et al., 2022), however under the name LES-EllipSys3D or $LES_{DTU}$.

The three inflow wind fields are generated using a bi-periodic precursor simulation of a pressure-driven isothermal boundary
layer. The computational domain extends $L_z = 1280$ m in the vertical direction, $L_x = 6L_z$ in the streamwise direction, and $L_y = 4L_z$ in the lateral direction. The grid is uniform in all coordinate direction, with $\Delta x = 20$ m and $\Delta y = \Delta z = 10$ m. A symmetry boundary condition is imposed at the domain top. At the surface, shear stress is prescribed using the Monin-Obukhov similarity theory (Monin, 1954) and the local instantaneous velocity sampled at the first grid point above the boundary. Inflow data for the main LES-ALM simulation, which are also employed by the DWM models, are extracted after a spin-up time of
30 000 s.

The domain of the $LES_{UU}$ simulation (i.e., including wind turbines) has the same dimensions $L_{x,y,z}$ as the precursor field. The inlet is located $6D$ upstream of turbine 1. In the turbine and wake region, the grid is uniform with a resolution of $\Delta x = \Delta y = \Delta z = D/32 = 3.9375$ m, starting $3D$ upstream of turbine 1 and extending $33D$ in the streamwise direction, and $4D$ in both the lateral and vertical directions. Outside this inner region, the grid is smoothly stretched towards the boundaries.
The turbine rotors are represented using ALMs (Sørensen and Shen, 2002), with each blade discretized into 32 elements. The ALM body forces are projected onto the grid with a three-dimensional Gaussian smearing function of width $\epsilon = 2\Delta x$. To mitigate spurious induction effects arising from the finite core-size of root and tip vortices, the smearing correction proposed by Meyer Forsting et al. (2019) is applied. Following a spin-up of 30 min, the main simulation is run for 45 min.

## 2.4 Wake tracking

From the flow field generated by $LES_{UU}$, the wake centre positions were tracked using the python toolbox SAMWICh developed at NREL. The wake centres were identified in the plane 5D downstream of each turbine normal to the wind-direction, for each time step using the two-dimensional Gaussian fit method (Trujillo et al., 2011) as implemented in the SAMWICh toolbox. To minimize algorithm error, the search area was limited to $\pm 1.25D$ of the turbine location laterally, and between $-0.5D$ and $D$ relative to the hub height vertically. After the wake centre time series were obtained for each turbine and downstream
location, four post-processing steps were applied to reduce error in the wake centre estimates. These post processing steps were determined based on a separate analysis conducted on $DWM_{NREL}$ simulation results, where SAMWICh wake centre detections were compared to actual wake centre values output directly from $DWM_{NREL}$.

1. Edge Detection Removal: wake centres detected at the search area edge were discarded and filled back in with linear interpolation.





2. Spike Removal 1: a median filter with kernel size of 15 seconds was applied to remove spurious spikes in the wake centre time series.

3. Jump removal: To remove remaining jumps in the wake centre time series, a rolling mean was applied on segments starting 20 seconds before the first, and 20 seconds after the last consecutive points, exceeding the maximum allowable gradient of $0.2D/s$.

4. Spike Removal 2: A final step identical to step 2 ensured that any spikes introduced by previous steps, primarily due to the arbitrary length of segment selected in step 3, were reduced from the final, post-processed wake centre time series.

Despite the improvements seen after post-processing the raw wake centres, the SAMWICh centres did still at times differ from the centres computed by $DWM_{NREL}$. This could happen because SAMWICh was tracking the aggregate deficit, made up of more than one wake deeper into the farm. The difference between the standard deviation of the the wake centre time series tracked by SAMWICh and the one obtained directly from $DWM_{NREL}$ stayed below $0.06D$ (i.e. 6 % of the rotor diameter). This value considers all inflow cases, and both lateral and vertical wake centre coordinates.

## 3 Results

### 3.1 Fully waked cases with varying ambient turbulence

In this section, we present a detailed comparison of the three DWM-based models under fully waked conditions for a row of four turbines exposed to aligned inflow. Three cases corresponding to low, medium, and high ambient turbulence conditions are considered, while maintaining below-rated wind speeds. We assess time-averaged flow fields, wake centre positions, power production, thrust forces, blade loads, and fatigue to identify key differences between the models and examine the influence of sub-modelling strategies.

### 3.1.1 Mean velocity profiles

Figure 1 shows time-averaged velocity profiles at $-1D$, $2.5D$, and $5D$ relative to the 4 turbine's streamwise positions for the low ambient turbulence case ($TI_a$ = 4.6 %). The upper row shows horizontal profiles at hub height and the lower row shows vertical profiles at the turbine's lateral centre. Horizontal dashed lines indicate the range of the turbine rotor's swept area. In the near-wake of turbine 1, at $x_{t=1} = 2.5D$, all the models except $DWM_{IFE}$ show velocity profiles with two minima reflecting the rotor thrust distribution. For $DWM_{DTU}$, this characteristic near-wake profile is more distinct than the LES profile, while the opposite goes for $DWM_{NREL}$. Both $DWM_{DTU}$ and $DWM_{NREL}$ predict the initial velocity profile downstream of the turbine by the BEM model. $DWM_{IFE}$, on the other hand, assumes a Gaussian wake-deficit profile for all $x$ downstream of the turbine. For $LES_{UU}$, $DWM_{DTU}$, and $DWM_{NREL}$, the velocity-deficit profiles have reached a Gaussian-like shape at $x_{t=1} = 5D$. While all the models show similar shapes of the horizontal profiles at $x_{t=1} = 5D$ and $x_{t=2} = -D$, the vertical profile of $LES_{UU}$ differs in shape from the other models with relatively higher deficits at the lower part of the rotor span. For turbines 2–4, both



**Figure 1.** Time-averaged velocity profiles for the aligned incoming wind case with low ambient turbulence ($\mathrm{TI_a} = 4.6\,\%$). Horizontal dashed lines indicate the rotor swept area.

$\mathrm{DWM_{NREL}}$ and $\mathrm{DWM_{DTU}}$ estimate the transition from BEM to Gaussian shape later than $\mathrm{LES_{UU}}$ which shows a Gaussian shape already at $x = 2.5D$. While the DWM models show symmetric horizontal velocity-deficits for the developed profiles, the $\mathrm{LES_{UU}}$ deficit has its maximum at $y < 0$. This small asymmetry for $\mathrm{LES_{UU}}$, which becomes more pronounced for the higher ambient turbulence cases, will be discussed in Sect. 3.1.2 where plots showing wake centre positions are presented. At the wake centre, $\mathrm{DWM_{IFE}}$ in general tends to under-predict the deficit slightly compared to $\mathrm{LES_{UU}}$. $\mathrm{DWM_{NREL}}$ shows good agreement to $\mathrm{LES_{UU}}$ at the wake centre at $x = 5D$ and $x = -D$, while $\mathrm{DWM_{DTU}}$ tends to slightly over-predict the wake centre deficit at these positions.

$\mathrm{DWM_{DTU}}$ and $\mathrm{DWM_{NREL}}$ show minor differences comparing the flow downstream of turbines 2–4 with turbine 1. The wakes of $\mathrm{DWM_{IFE}}$ and $\mathrm{LES_{UU}}$, however, show significant development as the deficit outside the rotor span increases along



**Figure 2.** Time-averaged velocity profiles for the aligned incoming wind case with medium ambient turbulence ($TI_a$ = 8.8 %). Horizontal dashed lines indicate the rotor swept area.

the row of turbines. Hence, $DWM_{IFE}$ and $LES_{UU}$ show lower gradients compared to $DWM_{DTU}$ and $DWM_{NREL}$ in the wake

shear layer between the wake-deficit and ambient for all turbines operating under waked conditions, and especially for turbine 4. The momentum conserving wake summation method applied in $DWM_{IFE}$, equation (4), seems to capture the impact of the wakes from far upstream, which has expanded over a long distance, but still the deficits are weaker to the sides and above the rotor span compared to $LES_{UU}$. The wake-deficit profiles predicted by $DWM_{DTU}$ and $DWM_{NREL}$, however, show less spreading. The maximum deficit operator in the $DWM_{DTU}$ model derives the turbine's incoming deficit at each radial position

as the smallest deficit scanning through the meandered deficits of all upstream turbines (see equation (2) and Larsen et al. (2013)). This seems to cause $DWM_{DTU}$ to predict only small variations in the incoming velocity fields for turbines 2–4, resulting in similar wakes. The fact that $DWM_{NREL}$ predicts only minor variations in the wake flow for turbines 2–4 is more



**Figure 3.** Time-averaged velocity profiles for the aligned incoming wind case with high ambient turbulence (TI$_a$ = 12 %). Horizontal dashed lines indicate the rotor swept area.

surprising given the sequential method for multiple wake handling applied by this DWM version (see equation (3)). However, the closer agreement of DWM$_{IFE}$ with LES$_{UU}$ in predicting wake development along the turbine row may also stem from the fact that DWM$_{IFE}$ is the only DWM implementation that incorporates a turbulence build-up model. This model accounts for the elevated incoming turbulence levels experienced by turbines 2–4 due to the added turbulence in the turbine wakes, leading to faster wake recovery through enhanced mixing and momentum entrainment from the ambient flow.

Figures 2 and 3 show time-averaged velocity profiles for the cases with medium (TI$_a$ = 8.8 %) and high (TI$_a$ = 12 %) ambient turbulence, respectively. The wake development behind each turbine is similar to the previous low-turbulent case, but due to higher turbulence levels and thus stronger meandering, the wakes show faster recovery and transition towards a Gaussian-like profile. As for the low-turbulence case, DWM$_{DTU}$ shows a more distinct near-wake profile than the other models at $x = 2.5D$



WIND
ENERGY
SCIENCE
DISCUSSIONS



**Figure 4.** Profiles of standard deviation of velocity for the aligned incoming wind case with low ambient turbulence ($\text{TI}_\text{a} = 4.6$ %). Horizontal dashed lines indicate the rotor swept area.

for all turbines for both $\text{TI}_\text{a} = 8.8$ % and $\text{TI}_\text{a} = 12$ %. For medium ambient turbulence conditions, only traces of the characteristic near-wake profile is visible in the wake of $\text{DWM}_\text{NREL}$ and $\text{LES}_\text{UU}$. For $\text{DWM}_\text{NREL}$ it is evident for $x = 2.5D$ downstream of all the turbines, while for $\text{LES}_\text{UU}$ it can be seen only downstream of turbine 1, at $x_{t=1} = 2.5D$. For the high ambient turbulence

case, the wakes predicted by both $\text{DWM}_\text{NREL}$ and $\text{LES}_\text{UU}$ have developed to a Gaussian profile at $x = 2.5D$ downstream of all turbines. However, it should be noted that for power and loads generation, the the near-wake at $x = 2.5D$ is of minor importance.

More relevant are the profiles at $x = 5D$ downstream and $x = -D$ just upstream the next turbine in the row. Here, $\text{DWM}_\text{IFE}$, and to a minor degree $\text{DWM}_\text{NREL}$, tend to under-predict the deficit at the wake centre, while $\text{DWM}_\text{DTU}$ slightly over-predicts

the deficit, compared to $\text{LES}_\text{UU}$. As for $\text{TI}_\text{a} = 4.6$ %, $\text{DWM}_\text{IFE}$ is the only DWM model capturing the increased deficit outside

**Figure 5.** Profiles of standard deviation of velocity for the aligned incoming wind case with medium ambient turbulence ($\mathrm{TI_a}$ = 8.8 %). Horizontal dashed lines indicate the rotor swept area.

the rotor span for the horizontal profiles along the turbine row, but not at the same level as $\mathrm{LES_{UU}}$. For the vertical profiles, however, no such increase is evident for $\mathrm{DWM_{IFE}}$ in the plots. Again, $\mathrm{DWM_{DTU}}$ and $\mathrm{DWM_{NREL}}$ show minor development in the flow along the turbine row, when comparing the wakes of turbine 1 and turbine 2, and especially when comparing the wakes of turbines 2–4.

$\mathrm{LES_{UU}}$ shows some notable differences in the flow field as the ambient turbulence level increase: as already mentioned, the asymmetry about $y = 0$ gets more pronounced for higher ambient turbulence, and strong acceleration of the flow is seen near the surface behind all turbines. In addition, the wake moves slightly away from ground, which is visible behind turbine 2 and further downstream. This is likely due to the non-zero turbine tilt angle for the $\mathrm{TI_a}$ = 8.8 % and $\mathrm{TI_a}$ = 12 % cases, causing the wakes to deflect upwards. Only the DWM model from NREL takes the turbine tilt into account when calculating the flow.







**Figure 6.** Profiles of standard deviation of velocity for the aligned incoming wind case with high ambient turbulence ($TI_a = 12\%$). Horizontal dashed lines indicate the rotor swept area.

Even though the upward wake deflection is not evident for $DWM_{NREL}$ in the velocity profiles, it becomes visible in the wake centre position plots in Sect. 3.1.2.

Figures 4-6 show profiles of axial velocity standard deviation, $\sigma_u$, for the three different ambient turbulence levels investigated. In general, $LES_{UU}$ shows much higher levels of $\sigma_u$ than the DWM models, with the exception for $DWM_{DTU}$ showing comparable levels to, and higher levels than, $LES_{UU}$ for the $TI_a = 8.8\%$ and $TI_a = 12\%$ cases, respectively. $DWM_{DTU}$ is the only DWM implementation with a wake added turbulence model applied in this study. For $TI_a = 4.6\%$ and $TI_a = 8.8\%$, the shapes of the $DWM_{DTU}$ profiles at $x_{t=1} = 2.5D$ are in good agreement to $LES_{UU}$ except close to the surface. In this regime, and also at $x_{t=1} = 5D$, the deviation from $LES_{UU}$ is likely not due to the wake added turbulence formulation itself, but because of the deviation in wake shape seen in Figs. 1 and 2, since the shape relates to the wake added turbulence formulation via the





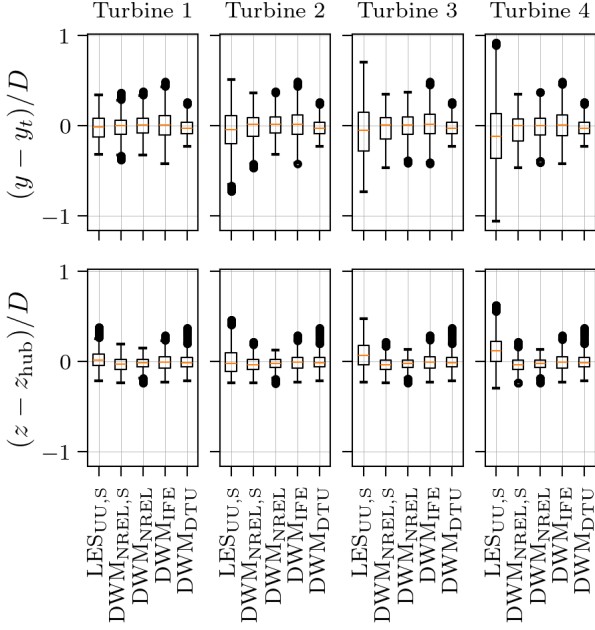

**Figure 7.** Box plot of horizontal (upper row) and vertical (lower row) wake centre position at $x = 5D$ behind the turbines, for the aligned incoming wind case with low ambient turbulence ($\text{TI}_\text{a}$ = 4.6 %).

velocity gradient. Deviation in predicted wake shape compared to $\text{LES}_\text{UU}$ also impact the levels of $\sigma_u$ for the two other DWM
models without a wake added turbulence model. This is due to wake meandering, causing regimes where the DWM models predict higher velocity gradients compared to $\text{LES}_\text{UU}$ to also experience higher variations in velocity as the wake meanders. This is highly visible around the lower part of the rotor's swept area in the wake of turbines 1 and 2 for the cases with higher ambient turbulence, $\text{TI}_\text{a}$ = 8.8 % and $\text{TI}_\text{a}$ = 12 %. Here the DWM models show higher vertical velocity gradients, and consequently higher levels of $\sigma_u$, compared to $\text{LES}_\text{UU}$ which shows an almost flat vertical velocity profile in this regime. Also
for the turbines deeper into the turbine row, the deviation in predicted wake shape can to a large extent explain the difference in $\sigma_u$ seen between the DWM models and $\text{LES}_\text{UU}$. However, it gets clear that the lack of a model for turbulence build-up, as addressed in Keck et al. (2015) and Branlard et al. (2024), amplifies the difference along the turbine row. Even though not including a wake added turbulence model, the $\text{DWM}_\text{IFE}$ implementation includes a model for turbulence build-up. This is evident for the case of low ambient turbulence shown in Fig. 4, where for $\text{DWM}_\text{IFE}$ the levels of $\sigma_u$ along the turbine row
increase and become closer to the levels of $\text{LES}_\text{UU}$ in the wake of turbine 4. The same increase is not seen in Figs. 5-6, possible due to the already high ambient turbulence in these cases, thus making the wake turbulence build-up relatively smaller.



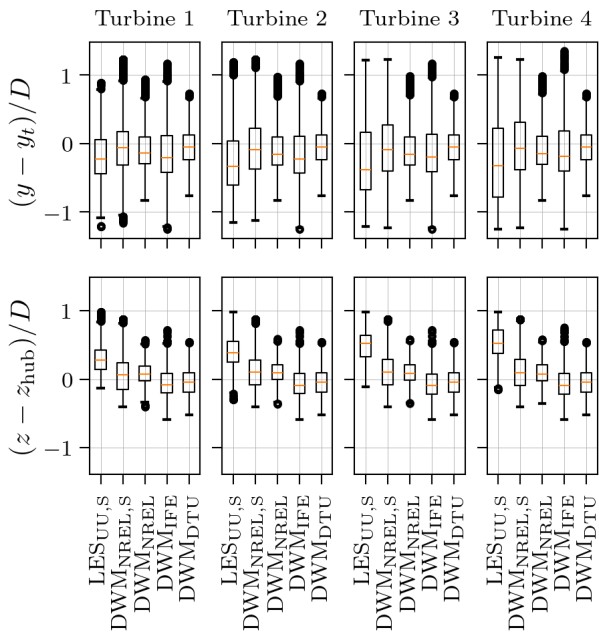

**Figure 8.** Box plot of horizontal (upper row) and vertical (lower row) wake centre position at $x = 5D$ behind the turbines, for the aligned incoming wind case with high ambient turbulence ($\mathrm{TI_a} = 12$ %).

### 3.1.2 Wake centre positions

Figures 7-8 show distributions of horizontal and vertical wake centre positions relative to the hub positions, at axial positions $5D$ downstream of each turbine in the row for the low and high ambient turbulence levels. The distributions are shown as box and whisker plots, where the box indicate the range of the first and third quartiles and the orange line shows the median wake centre position. Whiskers are extending to the most extreme, non-outlier data point, and outliers shown as circles are defined to be points located outside the box with more than 1.5 times the box size, $(1.5(Q_3 - Q_1))$. For the models marked with subscript S, the wake centre positions are tracked using the python toolbox SAMWICh developed at NREL, described in Sect. 2.4. For the rest of the models, the wake centre positions are taken directly from the meandering algorithm in the DWM simulations. For $\mathrm{DWM_{NREL}}$, both the wake centres from SAMWICh and directly from DWM are included in the figures to give an indication of the differences between the two approaches.

For $\mathrm{TI_a} = 4.6$ % all models predict the median wake centre position of turbine 1 to be around the hub, (0,0). For this low ambient turbulence case, the turbines are modelled with zero rotor tilt. The DWM models also predict the median positions of the wakes of turbines 2–4 to be close to (0,0), while the wake of $\mathrm{LES_{UU}}$ moves slightly upwards and to the right (negative $y$) deeper into the turbine row. The $\mathrm{TI_a} = 12$ % case shows similar results as $\mathrm{TI_a} = 4.6$ %, except that the wake of $\mathrm{LES_{UU}}$ moves further upwards, and $\mathrm{DWM_{NREL}}$ also moves slightly above hub height. A rotor-misalignment is well-known to deflect a wind turbine wake (Clayton and Filby, 1982), and the 5° tilt for the $\mathrm{TI_a} = 12$ % case is expected to deflect the wake upwards.





$\mathrm{DWM_{NREL}}$ is the only DWM model that take the effect of wake deflection due to tilt and yaw misalignment into account. For $\mathrm{TI_a}$ = 12 %, the median positions of the wake centres are clearly moving to the right (negative $y$) when looking downstream

for all models except $\mathrm{DWM_{DTU}}$, where the meandering as mentioned above is derived from a separate Mann box. The LES precursor has a small mean velocity component in $y$ direction of -0.23 $\mathrm{ms^{-1}}$ at hub position for the $\mathrm{TI_a}$ = 12 % case. If we assume that the wakes follow this velocity in $y$ direction as a passive tracer, the wake will have moved $\sim -17\mathrm{m} \sim -0.13D$ in $y$ direction at $5D$ downstream. This is in the same order of magnitude the DWM models predict the median wake centre position to be at $5D$ downstream for all the turbines, and can therefore explain the asymmetry seen for the DWM cases. With

the same reasoning, the wake centre should move $\sim -0.043D$ vertically due to a mean velocity component in $z$ direction of -0.076 $\mathrm{ms^{-1}}$. This is in the same order of magnitude as $\mathrm{DWM_{DTU}}$ and $\mathrm{DWM_{IFE}}$ predict the median wake centre position to be, while for $\mathrm{LES_{UU}}$ and $\mathrm{DWM_{NREL}}$ the upward deflection due to the rotor tilt has a larger contribution.

The asymmetry in the $\mathrm{LES_{UU}}$ deficit becomes more pronounced for the higher ambient turbulence case and deeper into the turbine row. Wake deflection for turbines with no misalignment between the rotor and the incoming wind has been observed

previously in both experiments (Bartl et al., 2018; Bossuyt et al., 2021) and LES (Fleming et al., 2014). As thoroughly explained by Zong and Porté-Agel (2020b) it follows from the streamwise momentum equation that for a counter-clockwise rotating wake in an undisturbed shear layer where the vertical velocity gradient is positive in the rotor area, the momentum balance will cause a wake deflection towards the left while the difference in tip vortex strength on the upper and lower part of the wake will have the opposite effect. These non-axisymmetric effects due to wake rotation and tip vortices are only captured by $\mathrm{LES_{UU}}$ and not

by the DWM models.

All models show more meandering in horizontal than vertical direction, and also increased meandering for higher $\mathrm{TI_a}$. For turbine 1 there is good agreement between the models in terms of meandering level. However, there is a tendency that $\mathrm{DWM_{IFE}}$ shows slightly more meandering than the other DWM models, with better match to LES for the low ambient turbulence case and slightly too high level of meandering at high ambient turbulence. According to Keck et al. (2013), a lower wake transport

velocity increases the level of meandering, which is in line with the higher meandering levels seen for $\mathrm{DWM_{IFE}}$ compared to $\mathrm{DWM_{DTU}}$.

For the wakes of turbines 2–4, the box plots show larger discrepancies between the models. While the wake of $\mathrm{LES_{UU}}$ shows $\sim 50$ % increase in spreading from turbine 1 to 2, there is no significant change for the DWM models. The wake meandering for $\mathrm{LES_{UU}}$ continue to increase from turbine 2 to 4. The constant wake meandering predicted by the DWM models for all

turbines is due to the meandering algorithm used by these models. DWM assumes that the wakes of all turbines are meandered by the same ambient turbulence field without wake added turbulence included. It is also worth noting that the wake tracking algorithm in the SAMWICh toolbox tracks the combined wake from all upstream turbines; for turbine 4 this means the sum of the wakes from turbine 1 to 4. Since the meandering of an isolated wake increases with downstream distance, the impact from the wakes of the upstream turbines, which have travelled further can possibly contribute to an increased meandering level of

the combined wakes downstream of turbines 2–4 tracked by SAMWICh. $\mathrm{DWM_{NREL,S}}$ shows a small increase in meandering for both $\mathrm{TI_a}$ = 4.6 % and $\mathrm{TI_a}$ = 12 % along the turbine row. We therefore cannot rule out that parts of the differences we see



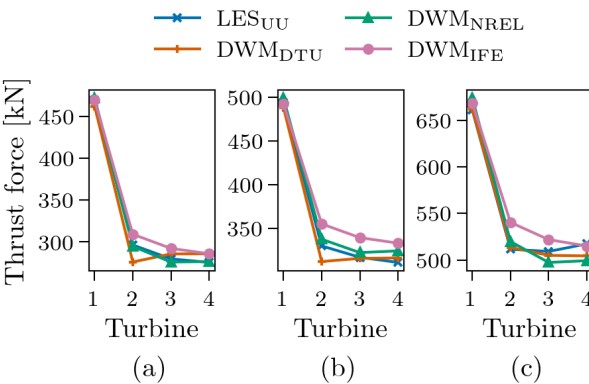

**Figure 9.** Mean thrust force for the aligned incoming wind case with (a) low ($TI_a$ = 4.6 %), (b) medium ($TI_a$ = 8.8 %), and (c) high ($TI_a$ = 12 %) ambient turbulence.

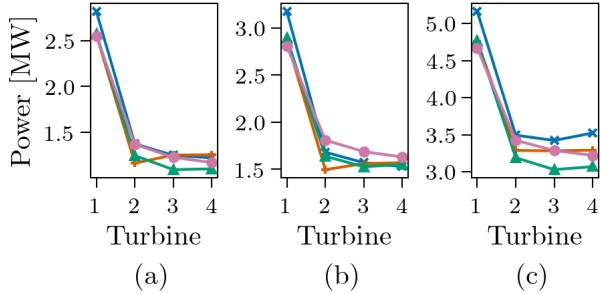

**Figure 10.** Mean power for the aligned incoming wind case with (a) low ($TI_a$ = 4.6 %), (b) medium ($TI_a$ = 8.8 %), and (c) high ($TI_a$ = 12 %) ambient turbulence. For legend, see Fig. 9.

in wake meandering between $LES_{UU}$ and the DWM models comes from the different methodologies used to identify the wake centres.

### 3.1.3 Power and thrust

Figures 9 and 10 show time-averaged thrust force and aerodynamic power for the three levels of ambient turbulence investigated. While all models show good agreement on the thrust force for turbine 1, $LES_{UU}$ shows about 10 % higher power compared to the DWM models for this turbine. As expected, there is a significant drop in both thrust and power for all models from turbine 1 to turbines 2–4 operating under waked conditions. For $DWM_{NREL}$, $DWM_{IFE}$, and $LES_{UU}$, both thrust and power decrease slightly from turbine 2 to 4 for the low and medium ambient turbulence cases. $DWM_{DTU}$ on the other hand shows a larger decrease from turbine 1 to 2 for both thrust and power compared to the two other DWM models, followed by an increase in power and thrust from turbine 2 to turbine 3 and 4. This effect has been seen in a row of turbines from full-scale





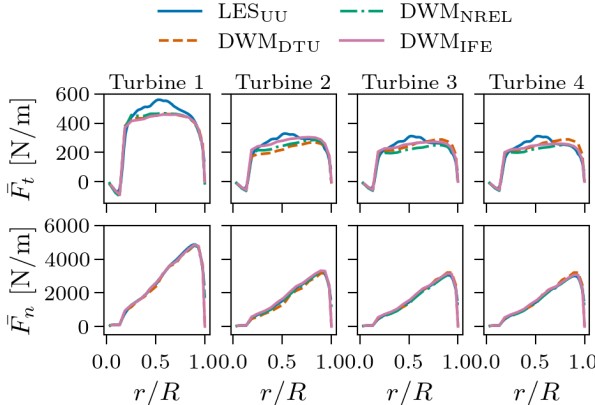

**Figure 11.** Time-averaged blade force as function of blade radius for the aligned incoming wind case with low ambient turbulence ($TI_a$ = 4.6 %).

measurements like e. g. the Lillgrund wind farm with similar inflow conditions (below-rated wind speeds and low ambient turbulence) but with the turbines more closely spaced (see e.g., Madsen et al., 2016). For $TI_a$ = 12 % the drop from turbine 1 to 2 is, as expected for higher ambient turbulence, smaller for all models due to faster wake recovery. From turbine 2 to 4, $DWM_{DTU}$ shows constant values of thrust and power, and not the increase as was seen for the low and medium ambient turbulence cases. $LES_{UU}$ on the other hand shows a small increase in thrust and power from turbine 3 to 4 for $TI_a$ = 12 %. This could be explained by a turbulence build-up along the turbine row accelerating wake recovery deeper into the farm.

### 3.1.4 Blade forces

The time-averaged tangential and normal force distribution along the radial positions of the blades in Fig. 11 show good agreement between the models. Results are shown for the low ambient turbulence case, but are qualitatively similar for the higher ambient turbulence cases (not shown here). The largest deviations are seen in the middle section of the blades. Here $LES_{UU}$ gives higher tangential forces compared to the DWM models, which explains the higher levels of power that was found in Fig. 10. These figures also reveal that for the $TI_a$ = 4.6 % and $TI_a$ = 8.8 % cases, where $DWM_{DTU}$ showed increased thrust and power for turbine 3 and 4 relative to turbine 2, both normal and tangential forces tend to be higher compared to the other models at the outer part of the blade ($r/R > 0.5$) for the two last turbines in the row. For the normal forces that are almost an order of magnitude larger than the tangential forces, the relative difference between the models are small.

Figures 12 and 13 show the azimuthal variation of the normal component of the blade force at four radial positions along the blade for the low and high ambient turbulence cases, respectively. The time-averaged normal force at each radial position, $\bar{F}_n$ (given in Fig. 11), is subtracted from the azimuthally varying force, and finally normalized by $\bar{F}_n$, to show only the relative force variation the blade experiences over the rotation. $\bar{F}_n^\phi$ is azimuthally binned with $\Delta\phi = 12°$ over all blade rotations during the 45-min simulations. The maximum blade force occurs around $\phi = 0°$. This is when the blade points upwards, which



WIND
ENERGY
SCIENCE
DISCUSSIONS

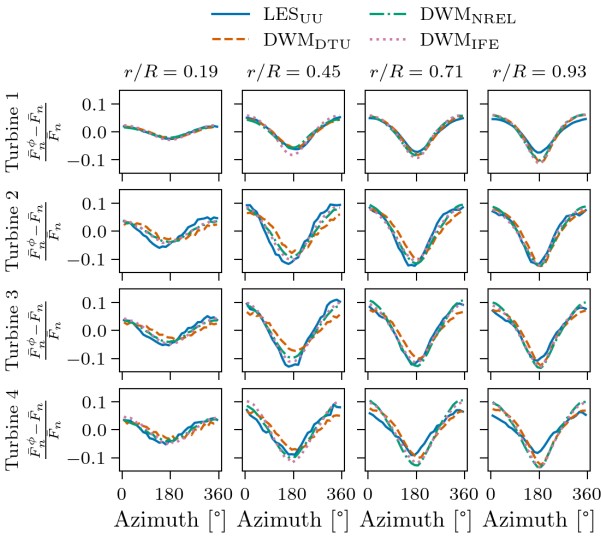

**Figure 12.** Relative difference between mean normal blade force per azimuthal bin $\bar{F}_n^\phi$ and total normal force $\bar{F}_n$, for the aligned incoming wind case with low ambient turbulence ($\text{TI}_a = 4.6$ %).

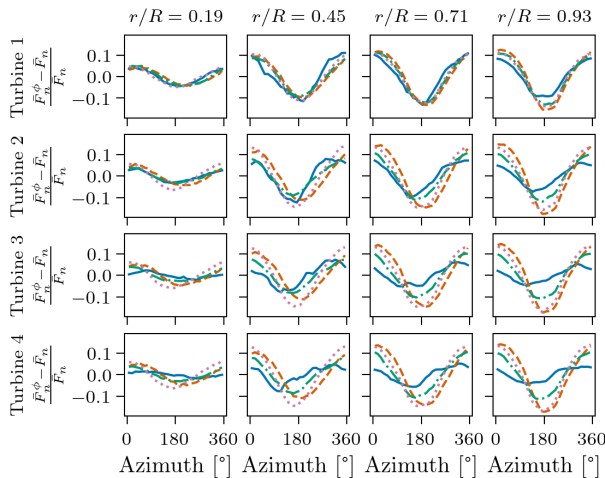

**Figure 13.** Relative difference between mean normal blade force per azimuthal bin $\bar{F}_n^\phi$ and total normal force $\bar{F}_n$, for the aligned incoming wind case with high ambient turbulence ($\text{TI}_a = 12$ %). For legend, see Fig. 12.

corresponds to when the blade experiences the highest wind speeds. Equivalently, the minimum blade force occurs around $\phi = 180°$. As mentioned in Sect. 2.1, the tower is not modelled in the simulations, and the difference in wind experienced by the blades are purely due to wind shear and, in addition for turbines 2–4, the influence from upstream wakes. All models show

increasing amplitudes in force variation towards the blade tip for all turbines.





For the low ambient turbulence case given in Fig. 12, the shape of the force variations shows good agreement between the models, even though the $\mathrm{DWM_{DTU}}$ force variations are slightly shifted towards higher $\phi$ for turbines 2–4 compared to the other models. Also the amplitudes of the force variations are generally in good agreement, however, $\mathrm{LES_{UU}}$ shows slightly smaller amplitudes than the DWM models at $r/R = 0.93$. Since turbine 1 experiences the same incoming wind profile for all models,

differences in force variations for this turbine are purely due to differences in the turbine aerodynamic models. $\mathrm{LES_{UU}}$ uses an ALM with a vortex-based tip/smearing correction by Meyer Forsting et al. (2019), while all the DWM models uses BEM with Prandtl tip correction (Glauert, 1935) for calculating blade forces (see Sect. 2.2.8). Different tip corrections are likely causing the deviations at $r/R = 0.93$.

For high ambient turbulence, Fig. 13, the deviations between the models are more prominent. For the turbines operating

under waked conditions, $\mathrm{LES_{UU}}$ exhibits a phase shift towards lower $\phi$ ($< 180°$), while $\mathrm{DWM_{DTU}}$ again shows a small shift towards higher $\phi$ ($> 180°$). For $\mathrm{LES_{UU}}$, the shift is largest for turbine 3 and 4, where the maxima have moved from $0°$ to $\sim 300°$, and the minima from $\sim 180°$ to $\sim 150°$. As discussed, the box plots in Fig. 8 show that the wakes move slightly to the right when looking downstream (to negative $y$). This also shifts the regime of highest wind to the left. Equivalently, the area of lowest wind will shift to the right, to $\phi < 180°$. As for the $\mathrm{TI_a} = 4.6$ % case, $\mathrm{LES_{UU}}$ predicts smaller force variations than the

DWM models at the blade tip for turbine 1. This supports that there are differences in the turbine aerodynamic models. For the turbines operating under waked conditions, the models show large deviations in amplitude. $\mathrm{LES_{UU}}$ shows smaller amplitudes in force variations compared to the DWM models, with $\mathrm{DWM_{NREL}}$ being closest. While the amplitudes of the force variations stay approximately constant for all turbines for $\mathrm{DWM_{DTU}}$ and $\mathrm{DWM_{IFE}}$, it is decreasing deeper into the turbine row for $\mathrm{LES_{UU}}$ and $\mathrm{DWM_{NREL}}$.

For turbines 2–4 operating under waked conditions, the varying incoming wind for the different models is the main source of the differences seen in the force variations. The amplitudes of the force variations depend on the variations in velocity the blades experience over a rotation. A vertical velocity profile with less variations over the rotor's swept area, as typically seen for $\mathrm{LES_{UU}}$ at $x_{t=3} = -D$ and $x_{t=4} = -D$ in Figs. 2 and 3, means a less force variations for the blades of turbine 3 and 4. Equivalently, the higher velocity gradients in the profiles estimated by $\mathrm{DWM_{DTU}}$ and $\mathrm{DWM_{IFE}}$ give larger amplitudes.

### 3.1.5 Fatigue

Figure 14 shows 45-min damage equivalent loads (DELs, see Sect. 2.1 for details) of blade root flapwise bending moment, where Wöhler coefficient of 10 is used for the blades. For the low ambient turbulence case, Fig. 14 (a), $\mathrm{LES_{UU}}$ shows a significant increase in DEL from turbine 1 to 2, followed by a constant level deeper into the turbine row. $\mathrm{DWM_{DTU}}$ is in very good agreement with $\mathrm{LES_{UU}}$ for this case, except for slightly higher DEL for turbine 2. $\mathrm{DWM_{NREL}}$ and $\mathrm{DWM_{IFE}}$ on the other

hand do not capture the increase in DELs from turbine 1 to 2, but gives DELs on the same level for all turbines.

Figure 14 shows that blade root flapwise bending moment DELs increase with higher $\mathrm{TI_a}$ for all the models. $\mathrm{DWM_{DTU}}$ shows a consistent development from turbine 1 to 4 for all $\mathrm{TI_a}$'s, with an increase from turbine 1 to 2, and a small decrease for the turbines further downstream. The other models have a different development for the higher $\mathrm{TI_a}$ cases compared to $\mathrm{TI_a}$ = 4.6 %. For $\mathrm{TI_a} = 8.8$ % and $\mathrm{TI_a} = 12$ %, $\mathrm{LES_{UU}}$ shows a considerable decrease (about 15–20 %) from turbine 1 to turbine





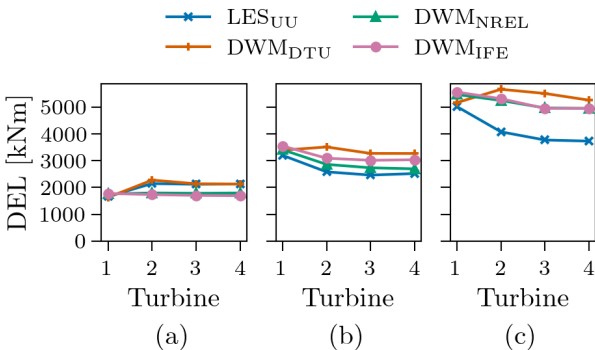

**Figure 14.** Fatigue of blade root flapwise bending moment for the aligned incoming wind case with ambient turbulence of (a) $\mathrm{TI_a}$ = 4.6 %, (b) $\mathrm{TI_a}$ = 8.8 %, and (c) $\mathrm{TI_a}$ = 12 %.

2, and then fairly constant levels from turbine 2 to 4. $\mathrm{DWM_{NREL}}$ and $\mathrm{DWM_{IFE}}$ show a similar development along the turbine row but with higher levels of DELs.

To understand the differences in DELs, we look at the power spectral density (PSD) of blade root flapwise bending moment presented in Fig. 15 as cumulative integrals. All models show jumps at $1P$ and higher harmonics, and also below $f_c$. $1P$ corresponds to the frequency of one blade revolution, and the jumps seen in the cumulative integral correspond to peaks in

a normal PSD. $f_c = U_\infty/2D$ is the meandering cut-off frequency, and loads associated to wake meandering are expected to appear below $f_c$ (Larsen et al., 2008; Larsen and Lio, 2025). Turbine 1, however, sees the undisturbed wind without a meandering wake, so for this turbine the energy levels in the PSD below $f_c$ can be seen as the baseline without any meandering energy. Surprisingly, only $\mathrm{DWM_{DTU}}$ shows any significant change in energy below $f_c$ between turbine 1 and the turbines operating under waked conditions. $\mathrm{LES_{UU}}$ do, however, show increased energy below $f_c$ for some cases, for example from

turbine 3 to 4 for the low ambient turbulence case. This change agrees well with the increased meandering seen in Fig. 7 especially in the lateral direction.

$\mathrm{DWM_{NREL}}$ and $\mathrm{DWM_{IFE}}$ shows fairly good agreement with $\mathrm{LES_{UU}}$ comparing the energy at $1P$ frequency. For these models the $1P$ energy levels seem to scale with the amplitude of the blade force variations in Figs. 12-13, with $\mathrm{DWM_{IFE}}$ having the largest and $\mathrm{LES_{UU}}$ the lowest amplitudes. $\mathrm{DWM_{DTU}}$, however, shows significantly higher $1P$ energy for the

turbines operating under waked conditions and especially for turbine 2 compared to the other models. For $\mathrm{DWM_{DTU}}$ the energy at $1P$ frequency does not scale as good with the amplitude of the blade force variations. For frequencies above $1P$, $\mathrm{LES_{UU}}$ shows increased levels of energy from turbine 1 to the turbines operating under waked conditions for the low ambient turbulence case, which likely come from the wake-added turbulence that is known to have a length scale considerably smaller than that of the ambient turbulence (Madsen et al., 2010), and is also seen as increased $\sigma_u$ in Fig. 4 from the inflow to the







**Figure 15.** Energy spectra of blade root flapwise bending moment for the aligned incoming wind case with ambient turbulence of (a) $TI_a$ = 4.6 %, (b) $TI_a$ = 8.8 %, and (c) $TI_a$ = 12 %.

waked conditions in front of turbines 2–4 [2]. None of the DWM models predict a significant increase in high-frequent energy for the turbines operating under waked conditions. Somewhat unexpectedly, this also holds for $DWM_{DTU}$, the only DWM implementation in this study that includes a wake-added turbulence model. For the higher ambient turbulence cases, $LES_{UU}$ shows no visible increase in high-frequency energy from turbine 1 to turbines 2–4. This, however, is likely due to the fact that the wake-added turbulence for these cases are negligible relative to the higher ambient turbulence levels.

---

[2]Unfortunately results from $LES_{UU}$ is not shown at $x_{t=i} = -D$, $i = 2, 3, 4$ in this case due to lack of time-resolved data at this axial position. However, $x_{t=i-1} = 5D$ is expected to show similar results.



WIND
ENERGY
SCIENCE
DISCUSSIONS

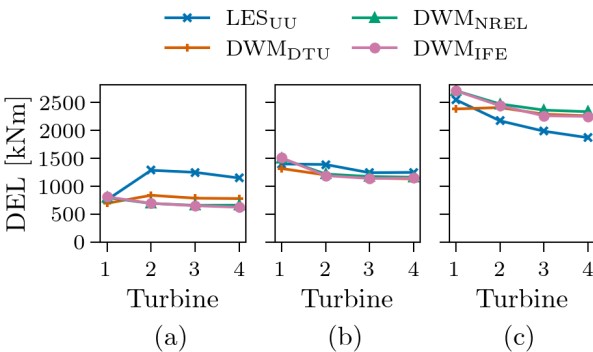

**Figure 16.** Fatigue of tower top yaw moment for the aligned incoming wind case with ambient turbulence of (a) $\text{TI}_\text{a}$ = 4.6 %, (b) $\text{TI}_\text{a}$ = 8.8 %, and (c) $\text{TI}_\text{a}$ = 12 %.

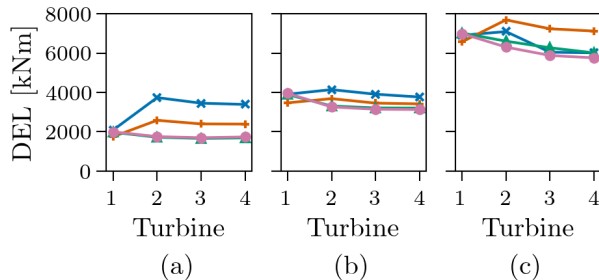

**Figure 17.** Fatigue of tower base fore-aft bending moment for the aligned incoming wind case with ambient turbulence of (a) $\text{TI}_\text{a}$ = 4.6 %, (b) $\text{TI}_\text{a}$ = 8.8 %, and (c) $\text{TI}_\text{a}$ = 12 %. For legend, see Fig. 16.

Even though the DELs estimated by $\text{DWM}_\text{DTU}$ is in good agreement with $\text{LES}_\text{UU}$ for the low ambient turbulence case, the underlying mechanisms for the development in DELs is rather different for the two models. The increase in DEL for $\text{DWM}_\text{DTU}$ from turbine 1 to turbines 2–4 is mainly due to increased energy content at $1P$ frequency but also below $f_c$ for all levels of ambient turbulence. For $\text{LES}_\text{UU}$ the increase in DEL for the low ambient turbulence case is due to a combination of increase in energy associated with $1P$ frequency and higher. For the higher ambient turbulence cases, the negligible wake-added turbulence

levels and decreasing energy content at $1P$ frequency for $\text{LES}_\text{UU}$ causes a reduction of DEL along the turbine row. For these cases $\text{DWM}_\text{NREL}$ and $\text{DWM}_\text{IFE}$ follows the same trend, and are closest to $\text{LES}_\text{UU}$.

    Figures 16 and 17 present 45-min DELs of tower-top yaw moment and tower-base fore-aft bending moment, respectively. A Wöhler coefficient of 3 is applied for the tower. For all cases, the models show good agreement in estimating tower DELs for turbine 1. For the low ambient turbulence case, $\text{LES}_\text{UU}$ predicts considerably higher DELs than the DWM models for the

downstream turbines. Among the DWM implementations, $\text{DWM}_\text{DTU}$ is closest to $\text{LES}_\text{UU}$, and is the only DWM model to reproduce the same development of DELs along the turbine row. At higher ambient turbulence levels, $\text{LES}_\text{UU}$ and the DWM





models are in closer agreement for the turbines operating under waked conditions. Consistent with the blade load results, tower loads increase with ambient turbulence intensity. The cumulative integrals of tower base fore-aft bending moment PSD in Fig. 18 show jumps below $f_c$ and at $3P$ frequency across all models. Additionally, $\text{LES}_{\text{UU}}$ shows energy at the harmonics of the $3P$ frequency, visible as small jumps at $6P$. The energy below $f_c$ contributes more strongly for tower loads than for blade loads shown in Fig. 15. Similar to the blade load results, $\text{DWM}_{\text{DTU}}$ is the only model consistently predicting higher energy below $f_c$ for turbines 2–4 compared to turbine 1. All models predict comparable energy content at $3P$ for turbine 1. However, $\text{LES}_{\text{UU}}$ is the only model that shows increased energy content at $3P$ and its harmonics for downstream turbines. This increase diminishes with rising ambient turbulence, and at $\text{TI}_{\text{a}} = 12\,\%$, all models predict similar energy content at $3P$ and higher frequencies for all turbines.

## 3.2 Partially waked case

This section extends the analysis to a more complex inflow scenario, introducing a small misalignment between the mean wind-direction and the turbine row. The resulting partial-wake configuration better reflects typical operational conditions in wind farms, where turbines are rarely aligned perfectly with the wind. The same ambient conditions as the medium ambient turbulence case in Sect. 3.1 are used, with a $5°$ offset angle between the mean wind-direction and the turbine row and no yaw misalignment between the rotors and the mean wind. We evaluate model performance in terms of time-averaged flow fields, wake centre positions, power production, thrust forces, blade loads, and fatigue. The results are used to further investigate each model's ability to capture asymmetric flow and loading.

### 3.2.1 Mean velocity profiles

Figure 19 shows time-averaged velocity profiles at three axial positions, at $-1D$, $2.5D$, and $5D$, relative to the 4 turbines for the partially waked case, with mean wind-direction $5°$ relative to the row of turbines, and ambient turbulence $\text{TI}_{\text{a}} = 8.8\,\%$. The upper row shows horizontal profiles at hub height and the lower row shows vertical profiles at the turbine's lateral centre. Horizontal dashed lines indicate the range of the turbine rotor's swept area of the closest upstream turbine, and horizontal dash-dot lines indicate at which lateral position the corresponding vertical profiles are plotted. Since the partially waked case has the same ambient conditions as the fully waked case with medium ambient turbulence, also the flow downstream and response of turbine 1 is similar to this case, with $\text{DWM}_{\text{DTU}}$ showing a distinct near-wake profile, and $\text{DWM}_{\text{NREL}}$ and $\text{LES}_{\text{UU}}$ only showing traces of the characteristic near-wake profile. As for the fully waked case, $\text{DWM}_{\text{IFE}}$, and to a minor degree $\text{DWM}_{\text{NREL}}$, tend to under-predict the deficit at the wake centre. $\text{DWM}_{\text{DTU}}$, however, slightly over-predicts the deficit, compared to $\text{LES}_{\text{UU}}$ in the wake of the first turbine, and then gradually under-predicts the deficit deeper into the turbine row. Do to the asymmetric conditions for turbines 2–4, all models estimate decreased wind speeds on the left side of the the wake when looking downstream for these turbines, but as for the fully waked case, $\text{LES}_{\text{UU}}$ estimates the deficit to the sides and above the rotor span to increase more than the DWM models deeper into the turbine row.

Figure 20 shows profiles of velocity standard deviation, $\sigma_u$, for the partially waked case. As for the fully waked case with medium ambient turbulence, $\text{LES}_{\text{UU}}$ and $\text{DWM}_{\text{DTU}}$ show much higher levels of $\sigma_u$ than $\text{DWM}_{\text{NREL}}$ and $\text{DWM}_{\text{IFE}}$.



**Figure 18.** Energy spectra of tower base fore-aft bending moment for the aligned incoming wind case with ambient turbulence of (a) $TI_a =$ 4.6 %, (b) $TI_a =$ 8.8 %, and (c) $TI_a =$ 12 %.

$DWM_{DTU}$ shows particularly good agreement with $LES_{UU}$ for the shear-layer to the right when looking downstream, while the left shear-layer and the vertical profile show larger differences.

### 3.2.2   Wake centre positions

Figure 21 shows box and whisker plots of horizontal and vertical wake centre positions $5D$ downstream of each turbine in the row for the partially waked case. For $LES_{UU,S}$, the wake centre positions are tracked using the python toolbox SAMWICh





**Figure 19.** Time-averaged velocity profiles for the partially waked case (5° inflow angle) with medium ambient turbulence (TI$_a$ = 8.8 %). Horizontal dashed lines indicate the rotor swept area of the closest upstream turbine, and horizontal dash-dot lines indicate at which lateral position the corresponding vertical profiles are plotted.

developed at NREL, described in Sect. 2.4. For the DWM models, the wake centre positions are taken directly from the meandering algorithm in the DWM simulation.

For turbine 1, all models predict that the median wake centre position has not moved away from the turbine position laterally, i.e. $y - y_t \approx 0$. For the DWM models this also true for turbines 2–4, while for LES$_{UU,S}$ the median wake centre position is positioned slightly to the left when looking downstream ($y - y_t > 0$). For the current case with skewed inflow, the turbines are positioned to the left of the their downstream turbines when looking downstream, e.g. turbine 1 is positioned 0.65D, 1.31D, and 1.96D to the left of turbine 2, 3, and 4, respectively. Since the wake tracking algorithm in the SAMWICh toolbox tracks the combined wake from all the upstream turbines, the shift towards left observed in the LES$_{UU,S}$ wake centre distribution,



**Figure 20.** Profiles of standard deviation of velocity for a wind-direction for the partially waked case (5° inflow angle) with medium ambient turbulence (TI$_a$ = 8.8 %). Horizontal dashed lines indicate the rotor swept area of the closest upstream turbine, and horizontal dash-dot lines indicate at which lateral position the corresponding vertical profiles are plotted.

can therefore possibly be a result of the impact from the upstream turbines, and not due to a deflection of the individual wakes. Alternatively, the asymmetric wake distribution can be explained by wake deflection due to the vertical velocity gradient in

the shear flow causing the wake to move horizontally, as discussed in Sect. 3.1.2. However, this does not explain why only the wakes of turbines 2–4 would deflect to the left, and not the wake of turbine 1.

As for the aligned cases investigated in Sect. 3.1, LES$_{UU}$ and to a smaller extent DWM$_{NREL}$ estimate that the wakes deflect upwards above hub height due to the 5° rotor tilt angle. Also in agreement with the previous results, all models show more meandering in horizontal than vertical direction. However, the large increase in meandering level deeper into the row which

was seen for LES$_{UU}$ in the aligned case, is not observed here.



WIND
ENERGY
SCIENCE
DISCUSSIONS

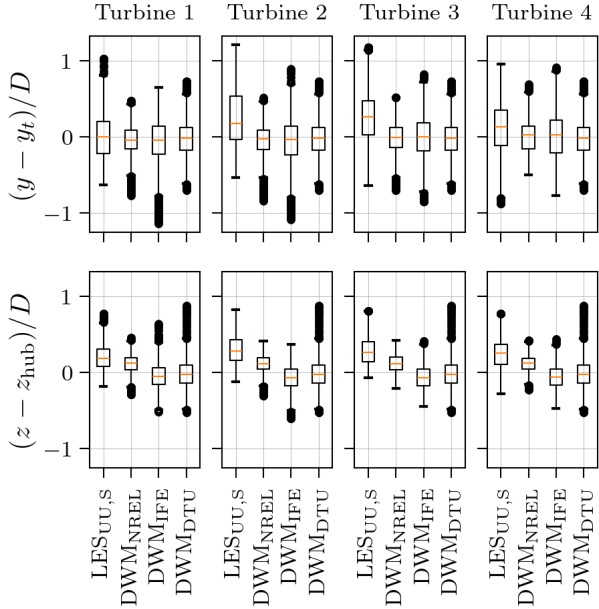

**Figure 21.** Box plot of horizontal (upper row) and vertical (lower row) wake centre position at $x = 5D$ behind the turbines for the partially waked case (5° inflow angle) with medium ambient turbulence ($TI_a$ = 8.8 %).

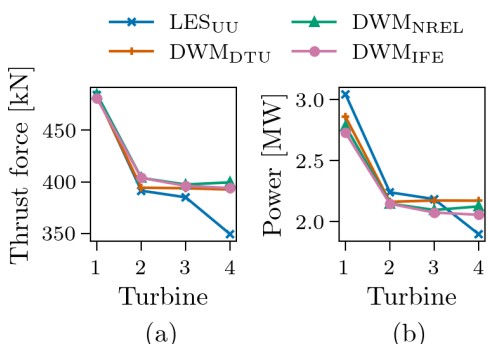

**Figure 22.** (a) Mean thrust force and (b) mean power for the partially waked case (5° inflow angle) with medium ambient turbulence ($TI_a$ = 8.8 %).

### 3.2.3 Power and thrust

Figure 22 shows time-averaged thrust force and power for the row of turbines for the partially waked case. As expected, all models show similar thrust and power as the fully waked case with the same medium ambient turbulence conditions for turbine 1. The small differences seen are due to variations in the incoming wind field at the two lateral positions of turbine 1 in the





fully waked case ($y$ =0) and in the partially waked case ($y \approx$123 m). As for the fully waked case, LES$_{\text{UU}}$ shows about 10 % higher power compared to the DWM models for turbine 1. For turbines 2–4 there are good agreement between the DWM models, while LES$_{\text{UU}}$ shows a significant drop in both thrust and power from turbine 3 to 4. This drop comes from the higher deficit upstream of turbine 4 as a result of a much wider horizontal wake estimated by LES$_{\text{UU}}$, seen at $x_{t=4} = -D$ in Fig. 19.

### 3.2.4 Blade forces

As for the fully waked case, there is good agreement between the models for all four turbines in the row when predicting time-averaged tangential and normal force distribution along the radial positions of the blades for the partially waked case (not shown here). In figure 23, the azimuthal variation of the normal blade force is presented at four radial positions along the blade for all four turbines in the row under the partially waked case. The models show larger deviations comparing the shapes of the force variations for turbines 2–4, operating under partially waked conditions. LES$_{\text{UU}}$ predicts the force minimum to be

shifted around 90° towards larger $\phi$ for these turbines, meaning the blades feel the lowest force when pointing straight to the left. The incoming wind field therefore has the lowest velocity towards the wake of the upstream turbine. For DWM$_{\text{IFE}}$ on the other hand, the minimum is shifted only slightly towards larger $\phi$. Due to the weaker wake-deficit estimated by this model, the blades feel the lowest force when pointing almost straight down where the velocity of the incoming flow is low due to shear. DWM$_{\text{DTU}}$ and DWM$_{\text{NREL}}$ shows minima located at the same $\phi$ as the LES$_{\text{UU}}$ minimum for all radial positions of the blade,

but towards the outer part of the blade a second minimum appears at approximately the same $\phi$ as for DWM$_{\text{IFE}}$. When the blades are pointing down, the tip enters into the regime where the DWM models estimate significantly sharper gradients in the vertical velocity profiles compared to LES$_{\text{UU}}$, as seen in Fig. 19. Consequently the DWM models predict a force minimum on the outer part of the blades at $\phi \approx$180° also for the turbines operating under partially waked conditions.

### 3.2.5 Fatigue

Figure 24 shows that all the models estimate comparable levels of blade and tower DELs for the partially waked case. DWM$_{\text{DTU}}$ is the only DWM model capturing a similar development along the turbine row as LES$_{\text{UU}}$, but at a slightly higher level for the blade DELs and lower levels for the tower DELs. DWM$_{\text{DTU}}$ and LES$_{\text{UU}}$ estimate an increase in DELs from turbine 1 to 2. The reduced loads due to an increased wind speed is compensated by increased turbulence downstream of turbine 1, modelled in DWM$_{\text{DTU}}$ by the wake-added turbulence model. DWM$_{\text{NREL}}$ and DWM$_{\text{IFE}}$, however, lack such a model, with the

result that the turbines operating under partially waked conditions show a decrease in DEL compared to turbine 1. By looking at the PSDs of the loads presented in Fig. 25, it is clear that DWM$_{\text{DTU}}$ also for the partially waked case shows more energy below $f_c$ than the other models for turbines 2–4. However, in contrast to the fully waked case, also LES$_{\text{UU}}$ shows increased energy below $f_c$ for turbines 2–4, for the blade root flapwise bending moment and tower base fore-aft bending moment. For the blade root flapwise bending moment, the DWM models estimate higher energy at the $1P$ frequency and its harmonics compared to

LES$_{\text{UU}}$ for all turbines, and is likely the main reason for the higher DELs estimated by the DWM models. Consistent with the fully waked case, the levels of energy at the $1P$ frequency coincides well with the difference in blade force variations seen in Fig. 23 for all models except DWM$_{\text{DTU}}$. Again DWM$_{\text{DTU}}$ shows a higher increase in $1P$ energy for the turbines operating





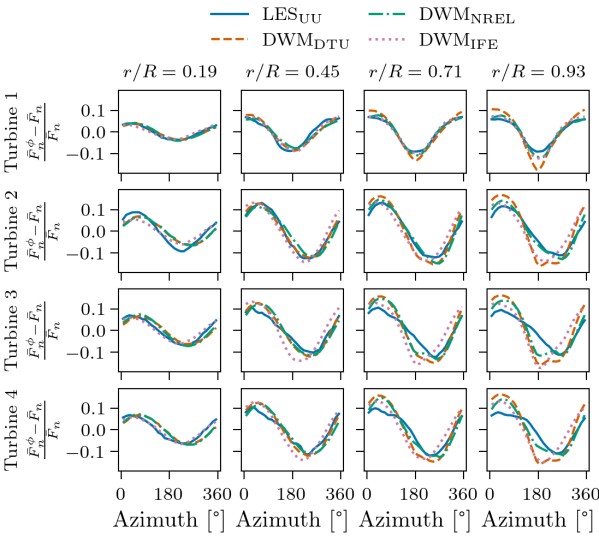

**Figure 23.** Relative difference between mean normal blade force per azimuthal bin $\bar{F}_n^\phi$ and total normal force $\bar{F}_n$ for the partially waked case (5° inflow angle) with medium ambient turbulence ($\mathrm{TI_a}$ = 8.8 %).

under waked conditions than the amplitude in blade force variations indicates. For tower-top yaw moment, $\mathrm{LES_{UU}}$ shows no significant change in energy below $f_c$, and the small decrease in $3P$ energy coincides well with the change in DELs along

the turbine row. $\mathrm{DWM_{NREL}}$ and $\mathrm{DWM_{IFE}}$ show a significant decrease in energy below $f_c$ for the turbines operating under waked conditions that together with a decrease in energy at $3P$ frequency reduce the tower-top yaw moment DELs. Lastly, $\mathrm{DWM_{DTU}}$ shows rather constant DELs as a result of a combination of increased energy below $f_c$ and decreased energy at $3P$ frequency for the turbines operating under waked conditions. For $\mathrm{DWM_{NREL}}$ and $\mathrm{DWM_{IFE}}$, the tower-base fore-aft bending moment DELs scale with the energy at $3P$ frequency, while for $\mathrm{LES_{UU}}$ and $\mathrm{DWM_{DTU}}$, it is the change in energy below $f_c$

that dominates the development in DELs along the turbine row.

## 4 Discussion

The comparative evaluation of the three DWM-based wake models against LES reveals generally good agreement in overall wake evolution and turbine performance trends, with notable discrepancies in specific wake features and load predictions.

### 4.1 Wake modelling

All three DWM models capture the qualitative shape and decay of the wake-deficits along the turbine row, but there are systematic differences in deficit magnitude and shape when compared to LES. Immediately downstream of the first turbine, the $\mathrm{DWM_{DTU}}$ model produces a more pronounced near-wake profile than observed in the LES, whereas $\mathrm{DWM_{NREL}}$ tends





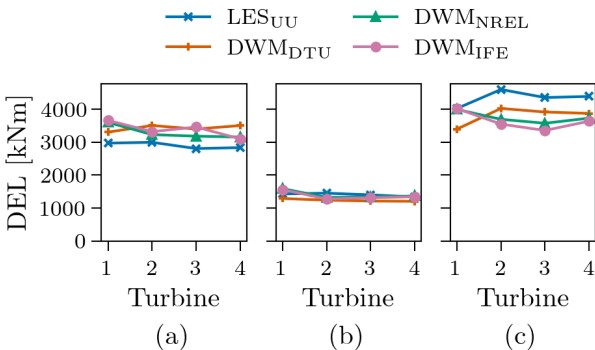

**Figure 24.** Fatigue of (a) blade root flapwise bending moment, (b) tower top yaw moment, and (c) tower base fore-aft bending moment for the partially waked case ($5°$ inflow angle) with medium ambient turbulence ($TI_a = 8.8$ %).

to produce a deficit more developed towards a Gaussian profile. For the turbines operating under waked conditions, $LES_{UU}$ already shows a Gaussian-like velocity profile at $x = 2.5D$. This is likely due to the added turbulence in the turbine wakes,

which increases the turbulence levels experienced by downstream turbines and enhances wake recovery through faster mixing. Neither $DWM_{DTU}$ nor $DWM_{NREL}$ reflect this change in wake recovery rate between turbine 1 and the turbines operating under waked conditions. Interestingly, $DWM_{IFE}$'s Gaussian profile therefore tend to outperform the other DWM models in the near-wake of the turbines operating under waked conditions. Although $DWM_{DTU}$ includes a wake-added turbulence model, it is only applied in the aeroelastic solver and does not influence the wake development. Consequently, increased wake recovery due

to elevated downstream turbulence is not captured in the velocity field. The newly implemented wake-added turbulence model in $DWM_{NREL}$ that couples wake-added turbulence and meandering (Branlard et al., 2024), though not applied in this study, may improve agreement in future comparisons. Nonetheless, both $DWM_{DTU}$ and $DWM_{NREL}$ exhibit faster wake recovery for higher ambient turbulence, as expected.

In the far-wake regions for the aligned case (e.g., $x_{t=i} = 5D$, $x_{t=i+1} = -D$), $DWM_{DTU}$ generally slightly overestimates

the centreline deficit, while $DWM_{IFE}$ slightly underestimates it. $DWM_{NREL}$ lies in between and is often closest to $LES_{UU}$ in these regions. However, for the partial wake case, $DWM_{DTU}$ seems to be closest to $LES_{UU}$. The $DWM_{DTU}$ model uses as mentioned previously a superposition method where, under below-rated conditions, the maximum velocity-deficit at each point is the maximum deficit scanning through all the individual meandered wake-deficits of upstream turbine is taken. In contrast, $DWM_{NREL}$ and $DWM_{IFE}$ incorporate wake summation schemes where all upstream wakes, calculated sequentially down the

row, affect the total flow field to different extent. While $DWM_{NREL}$ does not capture the change in flow field along the turbine row significantly better than $DWM_{DTU}$, $DWM_{IFE}$ demonstrates improved accuracy in the wake periphery, where the build-up of $LES_{UU}$ deficit is substantial. The build-up of TI along the turbine row in the $DWM_{IFE}$ model, affecting the eddy-viscosity closure, can also be a part of the difference observed relative to the other DWM models. The performance of $DWM_{IFE}$ does,



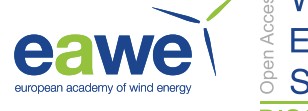

**Figure 25.** Energy spectra of (a) blade root flapwise bending moment, (b) tower top yaw moment, and (c) tower base fore-aft bending moment for the partially waked case (5° inflow angle) with medium ambient turbulence ($TI_a$ = 8.8 %).

however, degrade at higher ambient turbulence, suggesting that the term in the eddy viscosity model related to the ambient

wind shear scaling with turbulence intensity, should be calibrated.

Vertical velocity profiles play a critical role in load predictions as they affect the azimuthal variation of the inflow felt by turbine blades, influencing blade $1P$ and tower $3P$ loading. Rotor tilt induces an upward wake deflection, which is captured by $LES_{UU}$ and $DWM_{NREL}$ (the only model that incorporates tilt), but not by $DWM_{DTU}$ or $DWM_{IFE}$. Notably, $LES_{UU}$ predicts even greater upward deflection than $DWM_{NREL}$, which increases with downstream distance and occurs even for the low

turbulence case without rotor tilt. Also horizontal deflection is observed in the $LES_{UU}$ results. This suggests that wake rotation




and tip vortex effects, which are not accounted for in current DWM formulations, cause additional deflections. The curled wake model recently implemented in $\text{DWM}_{\text{NREL}}$ (Branlard et al., 2023), though not applied in this study, may improve agreement in future comparisons.

The box plots of wake centre positions show that the predicted meandering levels for the first turbine in the row agree
well among all DWM models and $\text{LES}_{\text{UU}}$. Wake meandering is consistently stronger in the horizontal than vertical direction across all models, consistent with large-scale vertical turbulence energy being less than large-scale lateral turbulence energy for conventional flat terrain conditions. However, the growth of meandering amplitude downstream is underrepresented: while $\text{LES}_{\text{UU}}$ shows a 50 % increase from turbine 1 to 2 (and continued growth thereafter), DWM models maintain nearly constant meandering levels. This might result from their reliance on the same ambient turbulence field for all wakes, without accounting
for the increased turbulence from upstream wake interactions, which leads to underprediction of wake spreading in deep arrays. Hanssen-Bauer et al. (2023) suggest that this could be addressed by coupling the wake-added turbulence model with the meandering routine, so that both ambient and wake-added turbulence contribute to wake motion. In fact, this is what is implemented in the new wake-added turbulence model in $\text{DWM}_{\text{NREL}}$ (Branlard et al., 2024). However, if the wake-added turbulence is in fact contributing significantly to wake meandering, it is in conflict with the traditional DWM assumption
that meandering is driven only by large-scale turbulence, while wake-added turbulence captures smaller scales. Nevertheless, important future work is to check this assumption by testing the new $\text{DWM}_{\text{NREL}}$ wake-added turbulence model.

As shown in the profiles of $\sigma_u$ (Figs. 4-6, and 20), the $\text{DWM}_{\text{DTU}}$'s inclusion of a wake-added turbulence model clearly improves its turbulence predictions. $\text{DWM}_{\text{NREL}}$ and $\text{DWM}_{\text{IFE}}$, lacking such a scheme, significantly underpredict $\sigma_u$ in the turbine wakes across all cases. Even with a wake-added turbulence model, $\text{DWM}_{\text{DTU}}$ shows discrepancies relative to $\text{LES}_{\text{UU}}$,
which may stem partly from differences in wake shape influencing the added turbulence through the velocity gradient input. However, the lack of a model for turbulence build-up across the row of turbines is evident, especially for the case with low ambient turbulence. Here $\text{LES}_{\text{UU}}$ indicates increasing $\sigma_u$ along the row, even as the mean deficit and gradients decrease, which contradicts the wake-added turbulence formulation given in Eq. 7. A reformulation of the present added wake approach could be to consider Eq. 7 as a source term of wake added turbulence (which it is) and then combine it with an accumulation term
and a decay term. The DWM modelling improvement by Keck et al. (2015) should also be considered as it both includes the impact of the ambient vertical wind gradient on the eddy viscosity and models the build-up of the wake added turbulence. Comparisons with ALM simulations show a reduction of up to 40 % in the deviation on turbulence intensity after the eighth turbine in a row by including this improvement (Keck et al., 2015).

## 4.2 Power and thrust predictions

Despite the differences in flow details, all three DWM-based models reproduce the general trends in time-averaged turbine power and thrust observed in the LES benchmark, and stay within 5–10 % deviation of the LES results. Surprisingly, while the DWM models show excellent agreement in power estimation for turbine 1 where the inflow is identical for all models, $\text{LES}_{\text{UU}}$ consistently predicts approximately 10 % higher power output. This comes from higher tangential forces in the middle sections of the blades estimated by $\text{LES}_{\text{UU}}$ compared to the DWM models. If this discrepancy seen in the turbine model is consistent





for all velocities, its impact can be adjusted for by normalizing all turbine powers with that of turbine 1. This would result
in the power from LES$_\mathrm{UU}$ generally align well with DWM$_\mathrm{NREL}$ for all turbines, while DWM$_\mathrm{DTU}$ and DWM$_\mathrm{IFE}$ typically
estimate slightly higher power outputs for turbines 2–4. However, the partially waked case with a 5° angle between the inflow
and the turbines row shows the consequence of the DWM models not capturing the significant build-up of velocity-deficit to
the sides of the rotor span. This build-up causes a drop in power from turbine 3 to turbine 4 in the LES$_\mathrm{UU}$ results which is not

captured by the DWM models. This highlights the importance of accurate prediction of lateral wake spreading, particularly
under real-world conditions, where perfect alignment is rare, as it can cause a non-negligible overshoot of the estimated annual
energy production of a wind farm.

### 4.3 Fatigue-load predictions

Fatigue-load predictions represent the area of greatest divergence between the DWM and LES results, underscoring the chal-

lenges involved in modelling wake-induced unsteady inflow conditions and their structural consequences. While all three
DWM models are able to reproduce the general trends in time-averaged loads (e.g. mean blade forces or mean thrust), their
predictions of damage equivalent loads (DELs) vary substantially, particularly for turbines operating under waked conditions.

Spectral analysis reveals that all DWM models tend to overestimate the energy content at the $1P$ frequency for the blade
loads. This has shown to be related to the higher estimated azimuthal blade force variations, which is a direct consequence

of the shape of the predicted wake velocity profiles. However, for some of the models, especially DWM$_\mathrm{DTU}$, the blade force
variation seen in Figs. 12-13 and 23, and the energy at $1P$ frequency do not scale as expected. As explained in Hanssen-Bauer
et al. (2023), this mismatch likely comes from a wake meandering effect. When upstream wakes meander, the wake moves
normal to the wind. This causes additional velocity gradients to appear when a wake covers only parts of the downstream
turbines' swept rotor area. These partially waked conditions caused by wake meandering last for several blade rotations since

the meandering motion is slower than the blade rotation, $f_c < 1P$. The effect will to a large extent not be visible in the blade
force variation plots as the forces are averaged in each azimuthal bin over the whole simulation. However, the plots of wake
centre positions reveal that DWM$_\mathrm{DTU}$ does not estimate higher levels of meandering than the other models. Still, the loads
spectra for DWM$_\mathrm{DTU}$ clearly show a significant increase in energy at the low frequencies associated with meandering for
the turbines operating under waked conditions, which is not seen for the other models. This suggests that for DWM$_\mathrm{DTU}$, the

meandering has a higher effect on the loads compared to the other models, possibly due to the more distinct deficits with
sharper radial gradients in DWM$_\mathrm{DTU}$'s case.

Consistent with prior findings for above-rated conditions in Hanssen-Bauer et al. (2023), we find that all DWM models
tend to underestimate the fatigue-loading on downstream turbines, especially on the tower, if important turbulence-generation
mechanisms are neglected. In the present below-rated cases, the DWM$_\mathrm{DTU}$ model with a wake-added turbulence model is the

only DWM implementation that roughly captures the increase in fatigue-damage for the turbines exposed to upstream wakes
for the low ambient turbulence case. However, the magnitude of the increase in tower loads remains under-predicted. For the
cases with higher ambient turbulence the trend is not so clear, but for the blade loads LES$_\mathrm{UU}$ shows a decrease in DEL's along
the turbine row, while DWM$_\mathrm{DTU}$ still estimates an increase in DEL's from turbine 1 to 2. The reason for this inconsistency



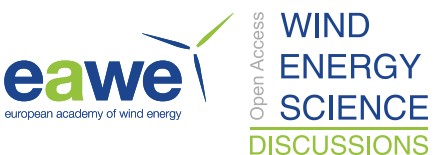

can be due to the different mechanisms behind the increased DEL's estimated by DWM$_{\text{DTU}}$ and LES$_{\text{UU}}$ for the turbines operating under waked conditions. For LES$_{\text{UU}}$, the high-frequency content of the load spectra is affected, probably directly by the higher small-scale turbulence in the wake. This is most pronounced in the case with low ambient turbulence, since the relative increase here is higher. DWM$_{\text{DTU}}$ seems to affect the loads more indirectly through wake meandering, as described in the previous paragraph. The wake-added turbulence can possibly play a role here, causing meandering of the wakes to have an increased effect on the loads when the wakes including enhanced turbulence move in and out of the rotor's swept area. For higher ambient turbulence, the wake-added turbulence seems to play a smaller role in the fatigue-loading development along the turbine row for LES$_{\text{UU}}$, while for DWM$_{\text{DTU}}$ the increased meandering effect on loading keeps dominating.

Another important factor explaining the divergence between the DWM and LES fatigue results is the fact that the DWM models predict axisymmetric wakes that meander, whereas the instantaneous LES wakes can be highly asymmetric and only approximately axisymmetric on average (as seen in velocity profiles in Figs. 1-3 and 19). This deviation from symmetry introduced by the wake in the LES$_{\text{UU}}$ results is likely an important driver on the tower loads, and is likely to contribute to the underprediction of tower loads in the DWM models.

An interesting observation is that LES$_{\text{UU}}$, which showed no significant change in energy below $f_c$ for the fully waked case estimates increased energy below $f_c$ for the turbines operating under partially waked conditions. This was seen for the blade root flapwise bending moment and tower-base fore-aft bending moment, but not the tower-top yaw moment. Since the upstream wakes on average are positioned to the side of the turbines operating in partial wake, combined with the fact that there is most meandering in lateral direction, results relatively often in the case where the wake moves completely away from the downstream turbine sideways. This causes large changes in the blade root moment and the tower fore-aft moment to appear in the meandering time-scale ($f < f_c$).

Overall, all DWM models struggle to capture the full range and intensity of wake-driven loading observed in LES. Better representation of turbulence evolution and its interaction with wake dynamics is crucial to improve fatigue-load predictions in DWM frameworks.

### 4.4 Weaknesses of the present study

While the comparative analysis provides valuable insights into the performance of DWM-based wake models, several limitations of the present study must be acknowledged.

First, all simulations were conducted at a single below-rated wind speed with the turbines running at constant RPM and pitch. This constrains the generalizability of the findings to other operational regimes, particularly near rated or cut-out wind speeds, where aerodynamic and control responses differ significantly. At higher wind speeds, turbine control strategies such as blade pitch and generator torque regulation may alter wake characteristics and structural responses in ways that are not captured here.

Second, the inflow conditions in both the LES and DWM simulations assume a neutral atmospheric boundary layer over homogeneous terrain, without thermal stratification. In reality, wind farms operate under more complex atmospheric conditions, including stable and unstable stratification, wind veer, and heterogeneous surface roughness. These factors influence turbulence





intensity, wake deflection, and recovery, and may lead to larger discrepancies between engineering-fidelity models and field measurements.

Third, the modelled wind farm layout consists of a single row of four identical turbines with uniform spacing. While this configuration provides a controlled environment for model comparison, it lacks the complexity of real-world wind farms, where turbines are arranged in staggered rows or irregular layouts and are subject to multi-directional wake interactions. Moreover, the only non-aligned inflow condition tested involved a small $5°$ offset, which is modest relative to real-world offsets caused by wind-direction variability or wake steering control strategies. Larger inflow angles including turbine yaw misalignments could

lead to more complex wake dynamics that challenge current DWM formulations.

     Fourth, the simulation durations were finite, and some load and flow statistics may be affected by sampling limitations (Liew and Larsen, 2022).

     Finally, validation against field measurements was not part of this study. While high-fidelity LES-ALM provides a physically consistent and high-resolution reference, the simulations are not necessarily reflecting one-to-one full-scale measurements

(Asmuth et al., 2022; Sood et al., 2022). The actuator-line method used in the LES model—though widely accepted as a high-fidelity approach—introduces its own approximations. The method represents blades as line forces rather than resolving blade-resolved flow features, which limits its accuracy in modelling near-wake vorticity, dynamic stall, and fine-scale unsteadiness. Comparing model predictions against full-scale SCADA or lidar data would further strengthen the conclusions and reveal model limitations under real operational conditions.

Future work should address these limitations by considering a broader set of operating conditions, implementing variable atmospheric stability and wind shear, and evaluating model performance in more complex farm layouts. Enhanced turbulence modelling—including coupling of wake-added turbulence with meandering and more flexible wake-merging strategies—remains a key area for development in DWM frameworks. Ultimately, continued benchmarking against both LES and high-quality field data is essential to advance the reliability of engineering-fidelity wake models for design and certification.

# 5    Conclusions

This study presents a comprehensive comparison of three DWM-based wake models, the DTU, IFE, and NREL implementations, against high-fidelity LES for a row of wind turbines operating under below-rated wind conditions. The main findings indicate that all three engineering-fidelity models capture the general wake evolution and turbine performance with reasonable accuracy in terms of mean values. However, notable discrepancies arise in wake shape and unsteady load predictions. Specifi-

cally, the time-averaged turbine thrust force and aerodynamic power outputs predicted by the DWM models generally aligned with the LES benchmarks (often within 5–10 %), suggesting that the DWM framework is broadly reliable for wind farm energy yield estimation in below-rated wind speeds for a variety of turbulence intensities.

     Nonetheless, the models exhibit limitations in capturing wake shapes, unsteady wake dynamics, and cumulative effects observed in the LES, and their strengths and weaknesses vary between the different DWM implementations. Accurate modelling

of far-wake shape is particularly important, as it influences both power output and structural loads on downstream turbines. For





instance, the pronounced deficit build-up observed in the peripheral regions of the LES wake may significantly affect power estimation under partially waked conditions. In this regard, the IFE model's wake superposition approach and its treatment of turbulence build-up via an eddy viscosity formulation appear to outperform the other models. Another key observation is the presence of relatively small wake deflections in the LES, even without turbine yaw. These deflections, likely resulting from ro-
tor tilt and three-dimensional wake effects, are crucial for predicting blade loading on downstream turbines. The NREL model most closely replicates this behaviour, capturing the upward wake deflection. Moreover, accurate representation of the increased turbulence in the turbine wakes—both spatial variation and spectral content—as well as the progression of turbulence and meandering throughout the wind farm, is essential. Among the models, the DTU implementation, with its wake-added turbulence model, is closest to LES in capturing these turbulence characteristics. Of critical concern are the substantial dis-
crepancies in fatigue-relevant load predictions between the DWM models and the LES. While all the DWM implementations tend to under-predicted fatigue-damage on downstream turbines in low ambient turbulence and over-predict in high ambient turbulence, the DTU model is generally best at capturing the trend of varying loads along the row.

The implications for wind farm modelling and design are significant. On the positive side, DWM-based models offer a computationally efficient method for simulating wakes and can reasonably predict average flow deficits and power production
across many scenarios. This makes them well-suited for wind farm layout optimization, control strategy development, and operational assessments, where full LES remains infeasible. The general agreement in power and thrust predictions supports the use of these engineering-fidelity tools for estimating energy production and first-order loads under below-rated, aligned inflow conditions. However, discrepancies in wake modelling performance present potential risks. If applied without careful validation, some DWM models may underestimate fatigue-loads or overestimate downstream performance, potentially resulting in
non-conservative designs.

In conclusion, while DWM models strike a favourable balance between accuracy and computational cost, further refinement is necessary to support all aspects of wind farm design and certification with confidence. Continued development and calibration of these engineering-fidelity models, informed by high-fidelity benchmarks and field measurements, are essential. Addressing the identified shortcomings will enable future DWM-based models to more accurately represent complex wake
interactions, thus enhancing predictions of both energy yield and structural loads in large wind farms.

*Author contributions.* All authors: proposed the methodology, formal analysis, and investigation. ØWHB: simulated the test cases and submitted the $DWM_{IFE}$ results. PD: simulated the test cases and submitted the $DWM_{NREL}$ results. HAaM: simulated the test cases and submitted the $DWM_{DTU}$ results. HA: Produced the LES inflow wind fields, simulated the test cases and submitted the $LES_{UU}$ results. ØWHB: post-processed (except for the wake centre tracking) and visualized the data from the numerical models. PD: conducted the SAMWICh wake
tracking and post-processing. ØWHB: wrote the manuscript draft. All authors: reviewed and edited the paper.

*Competing interests.* The authors declare that they have no conflict of interest.



*Acknowledgements.* This work has been funded by the Norwegian Research Council, through the project NEXTFARM: Engineering speed modelling of realistic fatigue for all the individual turbines in wind parks by representative pre-calculations, Grant No. 281020.

This work was authored in part by the National Renewable Energy Laboratory for the U.S. Department of Energy (DOE) under Contract No. DE-AC36-08GO28308. The views expressed in the article do not necessarily represent the views of the DOE or the U.S. Government. The U.S. Government retains and the publisher, by accepting the article for publication, acknowledges that the U.S. Government retains a nonexclusive, paid-up, irrevocable, worldwide license to publish or reproduce the published form of this work, or allow others to do so, for U.S. Government purposes.





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
