# Peer review of "Comparison of three DWM-based wake models at below-rated wind speeds"

_Wind Energy Science, 2025_

## Referee Comment (RC2)

**Comparison** of three DWM-based wake models at below-rated wind speeds**

Øyvind Waage Hanssen-Bauer1, Paula Doubrawa2, Helge Aa. Madsen3, Henrik Asmuth4, Jason Jonkman2, Gunner C. Larsen3, Stefan Ivanell4, and Roy Stenbro1

**Correspondence:** Øyvind Waage Hanssen-Bauer (oyvind.hanssen-bauer@ife.no)

**Abstract.** Wind turbine wake models are essential tools for predicting power losses and structural loads in wind farms. Among them, the dynamic wake meandering (DWM) model, included as a recommended approach in the International Electrotechnical Commission design standard, is a widely used engineering-fidelity method that balances accuracy and computational cost. This study compares the performance of three DWM-based wake model implementations (from the Technical University of Denmark, the National Renewable Energy Laboratory, and the Institute for Energy Technology) under below-rated wind speed conditions. Model predictions of wake flow, power output, and structural loads for a four-turbine row are evaluated across different ambient turbulence levels and wind-direction misalignments, and compared against high-fidelity large-eddy simulation results. All three models captured the overall wake evolution and mean turbine performance with reasonable accuracy; predicted time-averaged thrust and power were typically within 5—10 % of the large-eddy simulation benchmark. However, notable differences emerged in wake structure and unsteady load predictions, with discrepancies increasing for turbines further downstream. These differences highlight the importance of modelling choices such as wake summation and turbulence treatment, which strongly influence power-deficit and fatigue-load predictions. Comparison with large-eddy simulations reveals the strengths and weaknesses of each approach, indicating where improvements are needed. In general, the findings suggest directions for refining DWM models and improving their fidelity, ultimately enabling more robust wake predictions for wind farm design and operation. 15

**1 Introduction**

The wind energy industry has undergone significant development in recent decades, evolving from isolated, low-efficiency turbines to large-scale, modern wind farms. In these farms, spatial constraints and the need to minimize infrastructure and maintenance costs often lead to farm layouts with tightly spaced turbines. This evolution has increased the focus on turbine—turbine interactions, as wake effects have been identified as a major contributor to energy losses and elevated structural loads throughout the farm.

<sup>1Institute for Energy Technology

<sup>2National Renewable Energy Laboratory

<sup>3Technical University of Denmark

[revised manuscript text omitted]

---

## Author Comment (AC1)

**Author Response to Referee Comments**

We thank both referees for their thorough, constructive, and helpful reviews. Below we address each comment point-by-point, and explain any changes which will be made in the revised manuscript.

**Referee #1**

**Comment 1 (Introduction, lines 17–18):** The referee noted that our opening sentences mischaracterized the history of wind energy by implying that large wind farms only emerged in "recent decades", whereas significant developments already took place in the 1970s and 1980s. We thank the referee for pointing this out, these lines are now revised.

Comment 2 (Methodology – Equation 1 notation): The referee observed an inconsistency in the notation for Equation 1. The referee suggested changing the summation to run from k=1 to K (using K for the total number of load ranges) and to define m. We agree with the referee's suggestion and have corrected the notation in Equation 1. We also added a definition of m in the text as the material's Wöhler exponent.

Comment 3 (Methodology – IFE model wake summation): The referee pointed out that in the IFE model, the superposition of wake velocity deficits is based on Zong *et al*. (momentum conserving superposition), whereas the summation of wake-added turbulence is done via a root-sum-square approach. We have updated the Methodology section to explicitly mention the differing superposition methods used in the IFE model, as recommended by the referee.

Comment 4 (Results – 3p frequency and PSD for tower yaw moment): The referee noted that our discussion of frequency content in rotor loads (lines 485–490) did not mention the higher harmonics of the 1p frequency, which is clearly visible in the LES power spectral density (PSD) and comes from asymmetric rotor loading. The referee suggested we mention this in the discussion and consider adding PSD plots for the tower yaw moment in the aligned inflow case to further illustrate this point. We agree that this is an important point to discuss. In the revised manuscript we have added a note highlighting that the LES results show an increase in 2P, 3P, 4P... for the turbines operating in waked conditions relative to turbine 1, whereas the DWM models do not capture this feature. We also included an additional figure in an Appendix showing the PSD of the tower yaw moment for the aligned case.

Comment 5 (Results – wording on wind speed vs. loads at line 588): The referee questioned the statement "The reduced loads due to an increased wind speed" in line 588 of the original manuscript. The referee is correct that this was a mistake. We have corrected that sentence in the Results section.

Comment 6 (Discussion – fully rigid tower limitation): The referee noted that our study assumes a fully rigid tower, whereas in reality tower load spectra are strongly influenced by the tower's structural eigenfrequencies and their harmonics. By using a rigid tower, we neglect any interaction between the DWM-induced loading and the tower's dynamic response. The referee suggested that we discuss this as a limitation, since it could affect the tower damage equivalent load (DEL) conclusions. As mentioned in the methodology section, this was done to get results comparable to the LES-ALM, but we agree with the referee's point that this is a limitation. In the revised Discussion section under limitations, we added that the turbine model in our simulations employed a rigid tower, which likely affects the absolute levels and spectral characteristics of the tower loads. We note that without the flexibility of the tower, certain resonance phenomena are not captured. As a result, the tower fatigue load predictions and comparisons must be viewed in light of this simplification.

**Comment 7 (Editorial suggestions):** The referee provided a list of minor editorial issues. We have fixed all of these issues in the manuscript.

We thank Referee #1 for the positive feedback and constructive comments, which have helped us improve the clarity and quality of the manuscript.

**Referee #2**

We also thank Referee #2 for a thorough review and constructive feedback. The referee raised several major points and additional suggestions, which we address in detail below:

Major Comment 1 (Language clarity and grammar): The referee found that many sentences in the manuscript were not clear, and at times the meaning was ambiguous. We have undertaken a thorough language revision of the manuscript. All specific instances pointed out by the referee (and others we identified) have been corrected to ensure that the intended meaning is clear throughout the manuscript.

Major Comment 2 (Conclusions section refocus): The referee recommended rewriting the Conclusions section. We agree with the referee's assessment and will rewrite the Conclusions section to emphasize the novel findings of this work rather than general DWM background. In the revised Conclusions, we will summarize the main insights from our comparisons, and we focus on the physical and modeling aspects, as per the referee's suggestion.

Major Comment 3 (Influence of different aeroelastic solvers): The referee noted that each DWM implementation was coupled with a different aeroelastic solver, and thus it is ambiguous whether the differences in fatigue load predictions are due solely to the DWM model differences or partially due to using different structural solvers. The referee asked for

a more explicit discussion of the possible influence of the aeroelastic solvers on the results, or clarification of how much solver-specific effects might play a role. This is an important point, and we have addressed it in the revised manuscript. We added a paragraph explicitly discussing the potential influence of the different aeroelastic solvers on the thrust, power and fatigue load outcomes. Even though aero-elastic solvers generally have shown reasonable agreement in other projects such as IEA Wind Tasks (rotor aerodynamics, TURBINIA, and OC3-OC7), we acknowledge in the revised text that parts of the differences in the turbine response may stem from the underlying aeroelastic modeling differences.

Since we tried to ensure similar inflow upstream of the leading turbine (turbine 1) across the models, the spread in values for turbine 1 might be a good estimate of the "uncertainty" introduced in the results by the different aeroelastic solvers. In the discussion, we include selected examples of power and DEL's where the turbine 2-4 values are normalized by values for turbine 1 to see how this affects the results.

Major Comment 4 (Attributing differences to specific sub-models – need for sensitivity tests): In the Introduction we stated that one aim was to examine how differences in submodeling strategies affect performance. The referee points out that in our study design, we compared the three DWM frameworks as complete packages, meaning multiple submodel differences are all entangled. This makes it hard to attribute outcome differences to any single factor, and the referee worries that some of our interpretations might be subjective. They recommend performing additional sensitivity studies where one submodel at a time is swapped or varied within a single framework, to isolate the impact of each modeling choice and strengthen the conclusions. We fully understand the referee's concern about attributing differences to individual sub-model choices. Ideally, one would perform controlled sensitivity tests as described. However, implementing such tests would require substantial additional work, including modifying the codes to interchange subcomponents between the different DWM frameworks and rerunning many simulations. Unfortunately, this was not feasible for us at this stage. Instead, we have taken the referee's advice into account by revising the Discussion section. We explicitly acknowledge that our inter-comparisons reflect the combined effect of many differing sub-models. We highlight this as an area for future work, suggesting that follow-up studies could perform the kind of isolated sub-model tests the referee described.

Major Comment 5 (Explanation of  $\sigma_u$ /U in lines 335–345): The referee disagreed with the explanation we provided in the original manuscript (lines 335–345) regarding why a steeper time-averaged velocity gradient in the lateral/vertical direction, combined with wake meandering, would result in higher streamwise velocity fluctuations. We agree with the

referee that the original explanation was not accurately formulated. It is not correct to refer to the wake shapes in Figs. 1 - 3 (which are time-averaged) to explain the differences in velocity fluctuations. It is rather the instantaneous wake shape that affects the velocity fluctuations. We have rewritten that part of the manuscript.

**Minor Comment 1 (Figure clarity – add grid lines):** The referee suggested adding grid lines to figures. We have implemented this suggestion. All relevant plots in the manuscript now include light grid lines in the background.

Minor Comment 2 (Instantaneous velocity contour plots): The referee recommended including a few instantaneous plots of the streamwise velocity to qualitatively show differences between the LES and the DWM model predictions. Figure 1 and 2 show time series of instantaneous velocity profiles 2.5D and 5.0D downstream of turbine 1, while figure 3 shows instantaneous velocity profiles at t = 100 s for the same positions as figure 1-3 in the manuscript. All examples are taken from the aligned incoming wind case with medium ambient turbulence (TIa = 8.8 %). Even though the inflow conditions for the DWM and LES simulations have been conditioned to be the same upstream of turbine 1, the inflow develops differently downstream in the DWM and LES simulations. Further, such plots include, in addition to the ABL turbulence, (multiple) wakes and stochastic wake dynamics coupled to the ABL turbulence. However, we agree that these plots can give some additional insight into the differences between the models since statistics can hide some important details. Therefore, we agree to put some examples of instantaneous velocity plots in an appendix.

**Minor Comment 3 (Figure aesthetics and readability):** The referee also encouraged us to generally improve the readability and aesthetics of the figures. We have taken steps to improve the figures' clarity and appearance.

Figure 1: Time series of instantaneous velocity profiles 2.5D downstream of turbine 1 for the aligned incoming wind case with medium ambient turbulence ( $TI_a = 8.8\%$ ).

Figure 2: Time series of instantaneous velocity profiles 5.0D downstream of turbine 1 for the aligned incoming wind case with medium ambient turbulence ( $TI_a = 8.8 \%$ ).

Figure 3: Instantaneous velocity profiles at t = 100 s for the aligned incoming wind case with medium ambient turbulence ( $TI_a = 8.8\%$ ).